# Automatic Perturbation Analysis for Scalable Certified Robustness and Beyond

**Kaidi Xu[1,*], Zhouxing Shi[2,*], Huan Zhang[3,*], Yihan Wang[2]**
**Kai-Wei Chang[3], Minlie Huang[4], Bhavya Kailkhura[5], Xue Lin[1], Cho-Jui Hsieh[3]**

[1]Northeastern University    [2]Tsinghua University    [3]UCLA
[4]DCST, THUAI, SKLits, BNRist, Tsinghua University    [5]Lawrence Livermore National Laboratory
{xu.kaid, xue.lin}@northeastern.edu, zhouxingshichn@gmail.com, huan@huan-zhang.com,
wangyihan617@gmail.com, kw@kwchang.net, aihuang@tsinghua.edu.cn,
kailkhura1@llnl.gov, chohsieh@cs.ucla.edu

*Kaidi Xu, Zhouxing Shi and Huan Zhang contributed equally*

## Abstract

Linear relaxation based perturbation analysis (LiRPA) for neural networks, which computes provable linear bounds of output neurons given a certain amount of input perturbation, has become a core component in robustness verification and certified defense. The majority of LiRPA-based methods focus on simple feed-forward networks and need particular manual derivations and implementations when extended to other architectures. In this paper, we develop an *automatic* framework to enable perturbation analysis on any neural network structures, by generalizing existing LiRPA algorithms such as CROWN to operate on general computational graphs. The flexibility, differentiability and ease of use of our framework allow us to obtain state-of-the-art results on LiRPA based certified defense for fairly complicated networks like DenseNet, ResNeXt and Transformer that are not supported by prior works. Our framework also enables *loss fusion*, a technique that significantly reduces the computational complexity of LiRPA for certified defense. For the first time, we demonstrate LiRPA based certified defense on Tiny ImageNet and Downscaled ImageNet where previous approaches cannot scale to due to the relatively large number of classes. Our work also yields an open-source library for the community to apply LiRPA to areas beyond adversarial robustness without much LiRPA expertise, e.g., we create a neural network with a provably flat optimization landscape by applying LiRPA to network parameters and considering perturbations on model weights. Our open source library is available at https://github.com/KaidiXu/auto_LiRPA.

## 1 Introduction

Bounding the range of a neural network outputs given a certain amount of input perturbation has become an important theme for neural network verification and certified adversarial defense [48, 31, 45, 57]. However, computing the exact bounds for output neurons is usually intractable [21]. Recent research studies have developed perturbation analysis bounds that are sound, computationally feasible, and relatively tight [48, 54, 40, 47, 38, 46]. For a neural network function $f(\mathbf{x}) \in \mathbb{R}$, to study its behaviour at $\mathbf{x}_0$ with bounded perturbation $\delta$ such that $\mathbf{x} = \mathbf{x}_0 + \delta \in \mathbb{S}$ (e.g., $\mathbb{S}$ is a $\ell_p$ norm ball around $\mathbf{x}_0$), these works provide two linear functions $\underline{f}(\mathbf{x}) := \underline{\mathbf{a}}^\top \mathbf{x} + \underline{\mathbf{b}}$ and $\overline{f}(\mathbf{x}) := \overline{\mathbf{a}}^\top \mathbf{x} + \overline{\mathbf{b}}$ that are guaranteed lower and upper bounds respectively for output neurons w.r.t. the input under perturbation: $\underline{f}(\mathbf{x}) \leq f(\mathbf{x}) \leq \overline{f}(\mathbf{x})$ ($\forall \mathbf{x} \in \mathbb{S}$). We refer to the technique used in these works as **Li**near **R**elaxation

based **P**erturbation **A**nalysis (**LiRPA**). CROWN [54] and DeepPoly [40] are two representative LiRPA algorithms. Beyond its usage in neural network verification and certified defense, LiRPA is capable to serve as a general toolbox to understand the behavior of deep neural networks (DNNs) within a predefined input region, and has been used for interpretation and explanation [24, 37].

To compute LiRPA bounds, the first step is to obtain linear relaxations of any non-linear units [54, 36] (e.g., activation functions) in a network. Then, these relaxations need to be "glued" together according to the network structure to obtain the final bounds. Early developments of LiRPA focused on feed-forward networks, and it has been extended to a few more complicated network structures for real-world applications. For example, Wong et al. [50] implemented LiRPA for convolutional ResNet on computer vision tasks; Zügner & Günnemann [59] extended [48] to graph convolutional networks; Ko et al. [24] and Shi et al. [37] extended CROWN [54] to recurrent neural networks and Transformers respectively. Unfortunately, each of these works extends LiRPA with an ad-hoc implementation that only works for specific network architecture. This is similar to the "pre-automatic differentiation" era where researchers have to implement gradient computation by themselves for their designed network structure. Since LiRPA is significantly more complicated than backpropagation, non-experts in neural network verification can find it challenging to understand and use LiRPA for their purpose.

Our paper takes a big leap towards making LiRPA a useful tool for general machine learning audience, by generalizing existing LiRPA algorithms to general computational graphs. Our framework is a superset of many existing works [49, 54, 47, 24, 37], and our automatic perturbation analysis algorithm is analogous to automatic differentiation. Our algorithm can compute LiRPA automatically for a given PyTorch model without manual derivation or implementation for the specific network architecture. Importantly, our LiRPA bounds are differentiable which allows efficient training of these bounds. In addition, our proposed framework enables the following contributions:

• The flexibility and ease-of-use of our framework allow us to easily obtain state-of-the-art certified defense and robustness verification results for fairly complicated networks, such as DenseNet, ResNeXt and Transfomer that are hardly supported in existing works due to tremendous efforts required for manual LiRPA implementation.

• We propose *loss fusion*, a technique that significantly reduces the computational complexity of LiPRA for certified defense. We demonstrate the first LiPRA-based certified defense training on Tiny ImageNet and Downscaled ImageNet [5], with a *two-magnitude improvement* on training efficiency.

• Our framework allows flexible perturbation specifications beyond $\ell_p$-balls. For example, we demonstrate a *dynamic programming* approach to concretize linear bounds under discrete perturbation of synonym-based word substitution in a sentiment analysis task.

• We showcase that LiRPA can be a *valuable tool beyond adversarial robustness*, by demonstrating how to create a neural network with a provably flat optimization landscape and revisit a popular hypothesis on generalization and the flatness of optimization landscape. This is enabled by our unified treatment and automatic derivation of LiRPA bounds for parameter space variables (model weights).

## 2  Background and Related Work

Giving certified lower and upper bounds for neural networks under input perturbations is the core problem in robustness verification of neural networks. Early works formulated robustness verification for ReLU networks as satisfiability modulo theory (SMT) and integer linear programming (ILP) problems [10, 21, 43], which are hardly feasible even for a MNIST-scale small network. Wong & Kolter [49] proposed to relax the verification problem with linear programming and investigated its dual solution. Many other works have independently discovered similar algorithms [8, 31, 38, 47, 54, 40, 46] in either primal or dual space which we refer to as linear relaxation based perturbation analysis (LiRPA). Recently, Salman et al. [36] unified these algorithms under the framework of convex relaxation. Among them, CROWN [54] and DeepPoly [40] achieve the tightest bound for efficient single neuron linear relaxation and are representative algorithms of LiRPA. Several further refinements for the LiRPA bounding process were also proposed recently, including using an optimizer to choose better linear bounds [7, 29], relaxing multiple neurons [39] or further tighten convex relaxations [42], but these methods typically involve much higher computational costs. The contribution of our work is to extend LiRPA to its most general form, and allow automatic derivation and computation for general network architectures. Additionally, our framework allows a general purpose perturbation analysis for any nodes in the graph and flexible perturbation specifications, not

Table 1: Table of Notations

| Symbol | Meanings | Symbol | Meanings |
|---|---|---|---|
| $i, j, k$ | Any node on a computational graph | $\mathbf{x}_i$ | Value of an independent node, typically model input or parameters. |
| $o$ | Output node on a computational graph | $\underline{\mathbf{h}}_i, \overline{\mathbf{h}}_i$ | Lower/upper bound of node $i$ respectively |
| $m(i)$ | In-degree of node $i$ | $\underline{\mathbf{W}}_i, \underline{\mathbf{b}}_i, \overline{\mathbf{W}}_i, \overline{\mathbf{b}}_i$ | Parameters of linear lower/upper bounds of node $i$ respectively |
| $u(i)$ | Set of predecessor nodes (inputs) of node $i$ | $\underline{\mathbf{A}}_i, \overline{\mathbf{A}}_i$ | Linear coefficients of $h_i(\mathbf{X})$ terms in the linear lower/upper bounds of $h_o(\mathbf{X})$ |
| $\mathbb{S}$ | The space of the perturbed input | $\underline{\mathbf{d}}, \overline{\mathbf{d}}$ | Bias terms in the linear lower/upper bounds of $h_o(\mathbf{X})$ during bound propagation |
| $\mathbf{X}$ | Concatenation of all $\mathbf{x}_i$ (assumed flattened) | $h_i(\mathbf{X})$ | Computed value of node $i$ on a computational graph |

limiting to perturbations on input nodes or $\ell_p$-ball perturbation specifications. This allows us to use LiRPA as a general tool beyond robustness verification.

The neural network verification problem can also be solved via many other techniques, for example, semidefinite programming [9, 33], bounding local or global Lipschitz constant [15, 33, 56]. However, LiRPA based verification methods typically scale much better than alternatives, and they are a keystone for many state-of-the-art certified defense methods. Certified adversarial defenses typically seek for a guaranteed upper bound on test error, which can be efficiently obtained using LiRPA bounds. By incorporating the bounds into the training process (which requires them to be efficient and differentiable), a network can become certifiably robust [48, 31, 45, 12, 55]. In addition, while interval bound propagation (IBP) [31, 12] that propagates constant bounding intervals can be easily extended to general computational graphs, bounds computed by IBP can be very loose and make stable training challenging [57]. Along with these methods, randomization based probabilistic defenses have been proposed [6, 28, 27, 35], but in this work we mostly focus on LiRPA based deterministic certified defense method.

Backpropagation [34] is a classic algorithm to compute the gradients of a complex error function. It can be applied automatically once the forward computation is defined, without manual derivation of gradients. It is essential in most deep learning frameworks, such as TensorFlow [1] and PyTorch [32]. The backward *bound* propagation in our framework is analogous to backpropagation as our computation is also automatic given the computational graph created by the forward propagation, but we aim to automatically derive bounds for output neurons instead of gradients. Our algorithm is significantly more complicated. On the other hand, LiRPA based bounds have been implemented manually in many previous works [49, 54, 45, 30], but they mostly focus on specific types of networks (e.g., feedforward or residual networks) for their empirical study, and do not have the flexibility to generalize to general computational graphs and irregular networks.

## 3  Algorithm

### 3.1  Framework of Perturbation Analysis on General computational Graphs

**Notations**   We define a computational graph as a Directed Acyclic Graph (DAG) $\mathbf{G} = (\mathbf{V}, \mathbf{E})$. $\mathbf{V} = \{1, 2, \cdots, n\}$ is a set of nodes in $\mathbf{G}$. $\mathbf{E}$ is a set of node pairs $(i, j)$ which denotes that node $i$ is an input argument of node $j$. For simplicity, we denote the in-degree of node $i$ as $m(i)$, and the set of input nodes for node $i$ as $u(i) = \{u_1(i), \cdots, u_{m(i)}(i)\}$ where $(u_j(i), i) \in \mathbf{E}, 1 \leq j \leq m(i)$. Each node $i$ has a few associated attributes: $H_i(\cdot)$ is the associated computation function, $\mathbf{h}_i = H_i(u(i))$ is the vector produced by node $i$. Although $\mathbf{h}_i$ can be a tensor in practice, we assume it has been flattened into a vector for simplicity in this paper. Each node $i$ is either an *independent node* with $m(i) = 0$ representing the input nodes of the graph (e.g., network parameters, model inputs), or a *dependent node* representing some computations (e.g., ReLU, MatMul). For independent nodes, $H_i$ is an identity function and we denote $\mathbf{h}_i = \mathbf{x}_i$. We let $\mathbf{X}$ be the concatenation of all $\mathbf{x}_i$, such that the output of each node $i$ can be written as a function of $\mathbf{X}$, $\mathbf{h}_i = h_i(\mathbf{X})$, without explicitly referring to $u_j(i)$. Without losing generality, we assume that the computational graph has a single output node $o$. To conduct perturbation analysis, we consider $\mathbf{x}_i$ to be arbitrarily taken from an *input space* $\mathbb{S}_i$. In particular, if $\mathbf{x}_i$ is not perturbed, $\mathbb{S}_i = \{\mathbf{c}_i\}$ and $\mathbf{c}_i$ is a constant vector. We denote $\mathbb{S}$ to be the space of $\mathbf{X}$ when each part of $\mathbf{X}$, i.e., $\mathbf{x}_i$, is perturbed within $\mathbb{S}_i$ respectively.

**Linear Relaxation based Perturbation Analysis (LiRPA)**   Our final goal is to compute provable lower and upper bounds for the value of output node $h_o(\mathbf{X})$, i.e., lower bound $\underline{\mathbf{h}}_o$ and upper bound $\overline{\mathbf{h}}_o$ (element-wise), when $\mathbf{X}$ is perturbed within $\mathbb{S}$: $\underline{\mathbf{h}}_o \leq h_o(\mathbf{X}) \leq \overline{\mathbf{h}}_o, \ \ \forall \mathbf{X} \in \mathbb{S}$. In LiRPA, we find tight lower and upper bounds by first computing linear bounds w.r.t. $\mathbf{X}$:

$$\underline{\mathbf{W}}_o \mathbf{X} + \underline{\mathbf{b}}_o \leq h_o(\mathbf{X}) \leq \overline{\mathbf{W}}_o \mathbf{X} + \overline{\mathbf{b}}_o \quad \forall \mathbf{X} \in \mathbb{S}, \tag{1}$$

---

**Algorithm 1** Forward Mode Bound Propagation on General Computational Graphs

---
    **function** BoundForward($i$)
      **for** $j \in u(i)$ **do**
        **if** attributes $\underline{\mathbf{W}}_j, \underline{\mathbf{b}}_j, \overline{\mathbf{W}}_j, \overline{\mathbf{b}}_j$ of node $j$ are unavailable **then**
          BoundForward(j)
      $(\underline{\mathbf{W}}_i, \underline{\mathbf{b}}_i, \overline{\mathbf{W}}_i, \overline{\mathbf{b}}_i) = G_i(\{B_j | j \in u(i)\})$

---

where $h_o(\mathbf{X})$ is bounded by linear functions of $\mathbf{X}$ with parameters $\underline{\mathbf{W}}_o, \underline{\mathbf{b}}_o, \overline{\mathbf{W}}_o, \overline{\mathbf{b}}_o$. We generalize existing LiRPA approaches into two categories: *forward mode* perturbation analysis and *backward mode* perturbation analysis. Both methods aim to obtain bounds (1) in different manners:

• **Forward mode**: forward mode LiRPA propagates the linear bounds of each node w.r.t. all the independent nodes, i.e., linear bounds w.r.t. $\mathbf{X}$, to its successor nodes in a forward manner, until reaching the *output node o*.

• **Backward mode**: backward mode LiRPA propagates the linear bounds of *output node o* w.r.t. *dependent nodes* to further predecessor nodes in a backward manner, until reaching all the *independent nodes*.

We describe these two different modes in details below.

**Forward Mode LiRPA on General Computation Graphs**   For each node $i$ on the graph, we compute the linear bounds of $h_i(\mathbf{X})$ w.r.t. all the independent nodes:

$$\underline{\mathbf{W}}_i\mathbf{X} + \underline{\mathbf{b}}_i \leq h_i(\mathbf{X}) \leq \overline{\mathbf{W}}_i\mathbf{X} + \overline{\mathbf{b}}_i \quad \forall \mathbf{X} \in \mathbb{S}.$$

We start from independent nodes. For an independent node $i$, we have $h_i(\mathbf{X}) = \mathbf{x}_i$ so we trivially have the bounds $\mathbf{I}\mathbf{x}_i \leq h_i(\mathbf{X}) \leq \mathbf{I}\mathbf{x}_i$. For a dependent node $i$, we have a *forward LiRPA oracle function $G_i$* which takes $\underline{\mathbf{W}}_j, \underline{\mathbf{b}}_j, \overline{\mathbf{W}}_j, \overline{\mathbf{b}}_j$ for every $j \in u(i)$ as input and produce new linear bounds for node $i$, assuming all node $j \in u(i)$ have been bounded:

$$(\underline{\mathbf{W}}_i, \underline{\mathbf{b}}_i, \overline{\mathbf{W}}_i, \overline{\mathbf{b}}_i) = G_i(\{B_j | j \in u(i)\}), \text{where } B_j := (\underline{\mathbf{W}}_j, \underline{\mathbf{b}}_j, \overline{\mathbf{W}}_j, \overline{\mathbf{b}}_j). \quad (2)$$

We defer the discussions on oracle function $G_i$ to a later section. Now, we focus on extending this method on a general graph with known oracle functions in Algorithm 1. The forward mode perturbation analysis is straightforward to extend to a general computational graph: for each dependent node $i$, we can obtain its bounds by recursively applying (2). We check every input node $j$ and compute the bounds of node $j$ if they are unavailable. We then use $G_i$ to obtain the linear bounds of node $i$. The correctness of this procedure is guaranteed by the property of $G_i$: given $B_j$ as inputs, it always produces valid bounds for node $i$. We analyze its complexity in Appendix A.2.

**Backward Mode LiRPA on General Computation Graphs**   For each node $i$, we maintain two attributes: $\underline{\mathbf{A}}_i$ and $\overline{\mathbf{A}}_i$, representing the coefficients in the linear bounds of $h_o(\mathbf{X})$ w.r.t $h_i(\mathbf{X})$:

$$\sum_{i \in \mathbf{V}} \underline{\mathbf{A}}_i h_i(\mathbf{X}) + \underline{\mathbf{d}} \leq h_o(\mathbf{X}) \leq \sum_{i \in \mathbf{V}} \overline{\mathbf{A}}_i h_i(\mathbf{X}) + \overline{\mathbf{d}} \quad \forall \mathbf{X} \in \mathbb{S}, \quad (3)$$

where $\underline{\mathbf{d}}, \overline{\mathbf{d}}$ are bias terms that are maintained in our algorithm. Suppose that the output dimension of node $i$ is $s_i$, then the shape of matrices $\underline{\mathbf{A}}_i$ and $\overline{\mathbf{A}}_i$ is $s_o \times s_i$. Initially, we trivially have

$$\underline{\mathbf{A}}_o = \overline{\mathbf{A}}_o = \mathbf{I}, \quad \underline{\mathbf{A}}_i = \overline{\mathbf{A}}_i = \mathbf{0}(i \neq o), \quad \underline{\mathbf{d}} = \overline{\mathbf{d}} = \mathbf{0}, \quad (4)$$

which makes (3) hold true. When node $i$ is a dependent node, we have a *backward LiRPA oracle function $F_i$* aiming to compute the lower bound of $\underline{\mathbf{A}}_i h_i(\mathbf{X})$ and the upper bound of $\overline{\mathbf{A}}_i h_i(\mathbf{X})$, and represent the bounds with linear functions of its predecessor nodes $u_1(i), u_2(i), \cdots, u_{m(i)}(i)$:

$$(\underline{\boldsymbol{\Lambda}}_{u_1(i)}, \overline{\boldsymbol{\Lambda}}_{u_1(i)}, \underline{\boldsymbol{\Lambda}}_{u_2(i)}, \overline{\boldsymbol{\Lambda}}_{u_2(i)}, \cdots, \underline{\boldsymbol{\Lambda}}_{u_{m(i)}(i)}, \overline{\boldsymbol{\Lambda}}_{u_{m(i)}(i)}, \underline{\boldsymbol{\Delta}}, \overline{\boldsymbol{\Delta}}) = F_i(\underline{\mathbf{A}}_i, \overline{\mathbf{A}}_i),$$

$$\text{s.t.} \quad \sum_{j \in u(i)} \underline{\boldsymbol{\Lambda}}_j h_j(\mathbf{X}) + \underline{\boldsymbol{\Delta}} \leq \underline{\mathbf{A}}_i h_i(\mathbf{X}), \quad \overline{\mathbf{A}}_i h_i(\mathbf{X}) \leq \sum_{j \in u(i)} \overline{\boldsymbol{\Lambda}}_j h_j(\mathbf{X}) + \overline{\boldsymbol{\Delta}}. \quad (5)$$

We substitute the $h_i(\mathbf{X})$ terms in (3) with the new bounds (5), and thereby these terms are backward propagated to the predecessor nodes and replaced by the $h_j(\mathbf{X})(j \in u(i))$ related terms in (5). In the end, all such terms are propagated to the independent nodes and $h_o(\mathbf{X})$ will be bounded by linear functions of independent nodes only, where (3) becomes equivalent to (1).

---

**Algorithm 2** Backward Mode Bound Propagation on a General Computational Graph

---

**function** BoundBackward($o$)

  Create BFS queue $Q$ and $Q.push(o)$

  $\underline{\mathbf{A}}_o \leftarrow \mathbf{I}, \ \overline{\mathbf{A}}_o \leftarrow \mathbf{I}, \ \underline{\mathbf{A}}_i \leftarrow \mathbf{0}, \ \overline{\mathbf{A}}_i \leftarrow \mathbf{0} \ (\forall i \neq o), \ \underline{\mathbf{d}} \leftarrow \mathbf{0}, \ \overline{\mathbf{d}} \leftarrow \mathbf{0}$  (Eq. (4))

  GetOutDegree($o$) {$\forall i$ obtain $d_i$, the number of unprocessed output nodes of node $i$ that $o$ depends on.}

  **while** $Q$ is not empty **do**

    $i \leftarrow Q.pop()$

    $(\underline{\mathbf{\Lambda}}_{u_1(i)}, \overline{\mathbf{\Lambda}}_{u_1(i)}, \underline{\mathbf{\Lambda}}_{u_2(i)}, \overline{\mathbf{\Lambda}}_{u_2(i)}, \cdots, \underline{\mathbf{\Lambda}}_{u_{m(i)}(i)}, \overline{\mathbf{\Lambda}}_{u_{m(i)}(i)}, \underline{\mathbf{\Delta}}, \overline{\mathbf{\Delta}}) = F_i(\underline{\mathbf{A}}_i, \overline{\mathbf{A}}_i)$  (Eq. (5))

    **for** $j \in u(i)$ **do**

      $\underline{\mathbf{A}}_j += \underline{\mathbf{\Lambda}}_j, \ \overline{\mathbf{A}}_j += \overline{\mathbf{\Lambda}}_j, \ d_j -= 1$

      **if** $d_j = 0$ and node $j$ is a dependent node **then**

        $Q.push(j)$

    $\underline{\mathbf{d}} += \underline{\mathbf{\Delta}}, \ \overline{\mathbf{d}} += \overline{\mathbf{\Delta}}, \ \underline{\mathbf{A}}_i \leftarrow \mathbf{0}, \ \overline{\mathbf{A}}_i \leftarrow \mathbf{0}$ {Clear $\underline{\mathbf{A}}_i$ and $\overline{\mathbf{A}}_i$ once we propagated through $i$.}

  **return** $\underline{\mathbf{d}}, \overline{\mathbf{d}}$ {The algorithm has modified $\underline{\mathbf{A}}_i, \overline{\mathbf{A}}_i$ on the graph.}

---

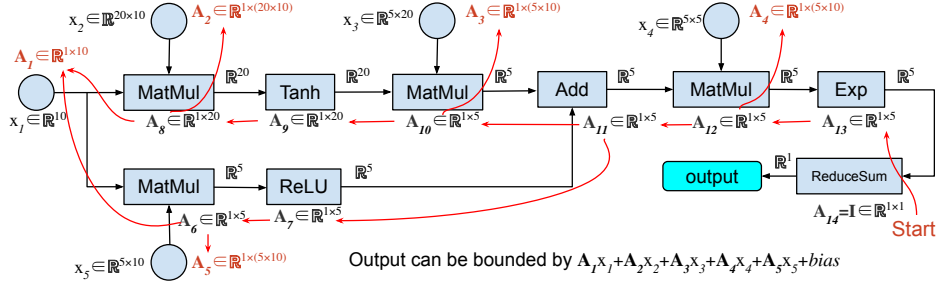

Figure 1: Illustration of the backward mode perturbation analysis. Node $1 \sim 5$ are independent nodes and the others are dependent nodes. Red arrows represent the flow of $\mathbf{A}$ matrices including both $\underline{\mathbf{A}}$ and $\overline{\mathbf{A}}$ that are propagated from the final output node (node 14) to previous nodes. Finally, only independent nodes retain non-zero $\mathbf{A}$ matrices (highlighted in red), and these matrices represent linear bounds w.r.t. independent nodes.

We present the full algorithm in Algorithm 2. We let $d_i$ denote the number of unprocessed output nodes of node $i$ that node $o$ depends on, which is initially obtained by a "GetOutDegree" function detailed in Appendix A.3. We use a BFS for propagating the linear bounds, starting from node $o$ as (4). For each node $i$ picked from the head of the queue, we backward propagate $h_i(\mathbf{X})$ using (5). We update the bound parameters and decrease all $d_j (j \in u(i))$ by one. If $d_j = 0$ becomes true for a dependent node $j$, all its related successor nodes have been processed and we push node $j$ to the queue. We repeat this process until the queue is empty. Figure 1 illustrates the flow of backward propagating the bound parameters on an example computational graph, and Figure 2 illustrates the BFS algorithm. We show its soundness in Theorem 1 and its proof is given in Appendix B.1.

**Theorem 1** (Soundness of backward mode LiRPA). *When Algorithm 2 terminates, we have*

$$\sum_{i \in \mathbf{V}} \underline{\mathbf{A}}_i h_i(\mathbf{X}) + \underline{\mathbf{d}} \leq h_o(\mathbf{X}) \leq \sum_{i \in \mathbf{V}} \overline{\mathbf{A}}_i h_i(\mathbf{X}) + \overline{\mathbf{d}} \quad \forall \mathbf{X} \in \mathbb{S},$$

*where $\underline{\mathbf{A}}_i, \overline{\mathbf{A}}_i$ are guaranteed to be $\mathbf{0}$ for any dependent node $i$, and thus we obtain provable linear upper and lower bounds of node $o$ w.r.t. all independent nodes.*

**Oracle Functions**    Oracle functions $F_i$ and $G_i$ are defined for each type of operations.[1] Previous works [48, 54, 36, 37] have covered many common operations such as affine transformations, activation functions, matrix multiplication, etc. Since the major focus of this paper is on handling general computational graph structures, rather than deriving bounds for these elementary operations, we leave the detailed form of these oracle functions in Appendix A.1.

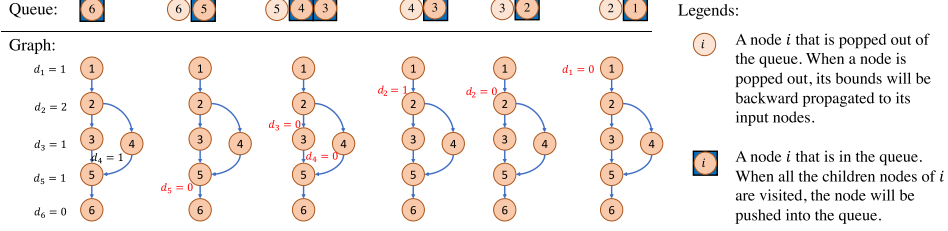

Figure 2: Flowchart of the BFS in Algorithm 2. In this example, node 6 is the final output node and $d_i$ is the number of unprocessed output nodes of node $i$ that node 6 depends on.

Some oracle functions depend on certain graph attributes. For example, $F_i$ of node $i$ with a nonlinear operation typically requires $\underline{\mathbf{h}}_j, \overline{\mathbf{h}}_j$ for all $j \in u(i)$ (typically referred to as "pre-activation bounds" in previous works). We can obtain $\underline{\mathbf{h}}_j, \overline{\mathbf{h}}_j$ by assuming node $j$ as the output node and apply Algorithm 2, then concretize the linear bounds as will be discussed in Sec 3.2. However, this can be very expensive because Algorithm 2 needs to be applied for every node $j$ wherever $\underline{\mathbf{h}}_j$ or $\overline{\mathbf{h}}_j$ is required, rather than just the output node. A typically more efficient approach is to obtain $\underline{\mathbf{h}}_j$ or $\overline{\mathbf{h}}_j$ for all dependent nodes except $o$ using a cheaper method and then apply backward mode LiRPA for node $o$ only. This leads to two variants of hybrid approaches, *Forward+Backward* and *IBP+Backward*, where $\underline{\mathbf{h}}_j$ and $\overline{\mathbf{h}}_j$ are produced by Forward Mode LiRPA or IBP, respectively. For certified training, *IBP+Backward* (generalized from *CROWN-IBP* in Zhang et al. [57]) is the best for efficiency. We discuss the time complexity of these methods in Appendix A.2.

## 3.2    General Perturbation Specifications and Bound Concretization

Once the linear bounds are obtained as (1), concrete bounds $\underline{\mathbf{h}}_o$ and $\overline{\mathbf{h}}_o$ can be found by solving the following optimization problems (this step is referred to as the "concretization" of linear bounds):

$$\underline{\mathbf{h}}_o = \min_{\mathbf{X} \in \mathbb{S}} \underline{\mathbf{W}}_o \mathbf{X} + \underline{\mathbf{b}}_o, \quad \overline{\mathbf{h}}_o = \max_{\mathbf{X} \in \mathbb{S}} \overline{\mathbf{W}}_o \mathbf{X} + \overline{\mathbf{b}}_o. \tag{6}$$

We show two examples: classic $\ell_p$-ball perturbations ($0 \le p \le \infty$), and synonym-based word substitution in language tasks.

$\ell_p$**-ball Perturbations**    In this setting, assuming that $\mathbf{X}_0$ is the clean input, the input space is defined by $\mathbb{S} = \{\mathbf{X} \mid \| \mathbf{X} - \mathbf{X}_0 \|_p \le \epsilon\}$ where the actual input $\mathbf{X}$ is perturbed within an $\ell_p$-ball centered at $\mathbf{X}_0$ with a radius of $\epsilon$. Linear bounds can be concretized as shown in Zhang et al. [54] for $0 < p \le \infty$:

$$\underline{\mathbf{h}}_o = -\epsilon \| \underline{\mathbf{W}}_o \|_q + \underline{\mathbf{W}}_o \mathbf{X}_0 + \underline{\mathbf{b}}_o, \quad \overline{\mathbf{h}}_o = \epsilon \| \overline{\mathbf{W}}_o \|_q + \overline{\mathbf{W}}_o \mathbf{X}_0 + \overline{\mathbf{b}}_o, \quad 1/p + 1/q = 1,$$

where $\| \cdot \|_q$ denotes taking the dual $\ell_q$-norm for each row in the matrix and the result is a vector. The case for $p = 0$ (sparse $\ell_0$ perturbations) is slightly different and will be discussed in Appendix C.4.

**Synonym-based Word Substitution**    Beyond $\ell_p$-ball perturbations, we show an example of a perturbation specification defined by synonym-based word substitution in language tasks. Let the clean input to the model be a sequence of words $w_1, w_2, \cdots, w_l$ mapped to embeddings $e(w_1), e(w_2), \cdots, e(w_l)$. Following a common adversarial perturbation setting in NLP [18, 20], we allow at most $\delta$ words to be replaced and each word $w_i$ can be replaced by words within its pre-defined substitution set $\mathbb{S}(w_i)$. $\mathbb{S}(w_i)$ is constructed from the synonyms of $w_i$ and validated with a language model. We denote each actual input word as $\hat{w}_i \in \{w_i\} \cup \mathbb{S}(w_i)$, and we show that the linear bounds of node $o$ can be concretized with dynamic programming (DP) in Theorem 2 as proved in Appendix B.2.

**Theorem 2.** *Let $\tilde{\underline{\mathbf{W}}}_t$ be columns in $\underline{\mathbf{W}}_o$ that correspond to the coefficients of $e(\hat{w}_t)$ in the linear bounds. The lower bound of $\underline{\mathbf{b}}_o + \sum_{t=1}^i \tilde{\underline{\mathbf{W}}}_t e(\hat{w}_t)$, when $j$ words among $\hat{w}_1, \ldots, \hat{w}_i$ have been replaced, denoted as $\underline{\mathbf{g}}_{i,j}$, can be computed by:*

$$\underline{\mathbf{g}}_{i,j} = \min(\underline{\mathbf{g}}_{i-1,j} + \tilde{\underline{\mathbf{W}}}_i e(w_i), \quad \underline{\mathbf{g}}_{i-1,j-1} + \min_{w'}\{\tilde{\underline{\mathbf{W}}}_i e(w')\}) \ (i,j > 0) \quad s.t. \ w' \in \mathbb{S}(w_i),$$

*and $\underline{\mathbf{g}}_{i,0} = \underline{\mathbf{b}}_o + \sum_{t=1}^i \tilde{\underline{\mathbf{W}}}_t e(w_t)$. The concrete lower bound is $\min_{j=0}^\delta \underline{\mathbf{g}}_{n,j}$. The upper bound can also be computed similarly by taking the maximum instead of the minimum in the above DP computation.*

### 3.3 *Loss Fusion* for Scalable Training of Certifiably Robust Neural Networks

The optimization problem of robust training can be formulated as minimizing the robust loss:

$$\min_\theta \sum_{\mathbf{X}_0, y} \max_{\mathbf{X} \in \mathbb{S}} L(f_\theta(\mathbf{X}), y), \tag{7}$$

where $f_\theta(\mathbf{X})$ is the network output at the logit layer, and $y$ is the ground truth. Let $g_\theta(\mathbf{X}, y) = (\mathbf{e}_y \mathbf{1}^\top - \mathbf{I}) f_\theta(\mathbf{X})$ be the margins between the ground truth label and all the classes (similarly defined in Wong & Kolter [49], Zhang et al. [57]). In previous works, the cross-entropy loss is upper bounded by lower bounds on margins, as a consequence of Theorem 2 in Wong & Kolter [49]: $\max_{\mathbf{X} \in \mathbb{S}} L(f_\theta(\mathbf{X}), y) \leq L(-\underline{g}_\theta(\mathbf{X}, y), y)$ where $\underline{g}_\theta(\mathbf{X}, y) \leq \min_{\mathbf{X} \in \mathbb{S}} g_\theta(\mathbf{X}, y)$. This requires us to first lower bound $g_\theta(\mathbf{X}, y)$ using LiRPA. The most efficient LiRPA approach [57] used IBP+Backward to obtain this bound, requiring $O(Kr)$ time where $K$ is the output (logit) layer size (equal to the number of classes in this case), and $O(r)$ is the time complexity of a regular computation pass without computing bounds (see Appendix A.2). This cannot scale to large datasets when $K$ is large (e.g. in Tiny ImageNet $K = 200$; in ImageNet $K = 1000$).

We propose a new technique, *loss fusion*, which computes an upper bound of $L(f_\theta(\mathbf{X}), y)$ directly without $\underline{g}_\theta(\mathbf{X}, y)$ as a surrogate. This is possible by treating $L$ as the output node of the computational graph. When $L$ is the cross entropy loss, we have $L(f_\theta(\mathbf{X}), y) = \log S(\mathbf{X}, y)$, where $S(\mathbf{X}, y) = \sum_{i \leq K} \exp([-g_\theta(\mathbf{X}, y)]_i)$. We can thus upper bound $L(f_\theta(\mathbf{X}), y)$ by computing an LiRPA upper bound for $S(\mathbf{X}, y)$ directly, yielding $\overline{S}(\mathbf{X}, y)$, and thereby $\max_{\mathbf{X} \in \mathbb{S}} L(f_\theta(\mathbf{X}), y) \leq \log \overline{S}(\mathbf{X}, y)$. This is a novel method that has not appeared in previous works and it yields two benefits. First, this reduces the time complexity of upper bounding $L(f_\theta(\mathbf{X}), y)$ to $O(r)$, as now the output layer size has been reduced from $K$ to 1. This is the first time in the literature that a tight LiRPA based bound can be computed in the *same asymptotic complexity as regular forward propagation* and IBP. Second, we show that this is not only faster, but also produces tighter bounds in certain cases:

**Theorem 3.** *Given same concrete lower and upper bounds of* $g_\theta(\mathbf{X}, y)$ *as* $\underline{g}_\theta(\mathbf{X}, y)$ *and* $\overline{g}_\theta(\mathbf{X}, y)$ *which may be used in linear relaxation, for* $S(\mathbf{X}, y) = \sum_{i \leq K} \exp([-g_\theta(\mathbf{X}, y)]_i)$, *we have*

$$\max_{\mathbf{X} \in \mathbb{S}} L(f_\theta(\mathbf{X}), y) \leq \log \overline{S}(\mathbf{X}, y) \leq L(-\underline{g}_\theta(\mathbf{X}, y), y), \tag{8}$$

*where $L$ is the cross-entropy loss, $\overline{S}(\mathbf{X}, y)$ is the upper bound of $S(\mathbf{X}, y)$ by backward mode LiRPA.* This theorem is proved in Appendix B.3. Intuitively, the original approach of propagating $\underline{g}_\theta(\mathbf{X}, y)$ through the cross-entropy loss is similar to using IBP for bounding the loss function, but in *loss fusion* we treat the loss function as part of the computational graph and apply LiRPA bounds to it directly; it produces tighter bounds as we can use a tighter relaxation for the nonlinear function $S(\mathbf{X}, y)$.

## 4  Experiments

Table 2: Error rates of different certifiably trained models on CIFAR-10 and Tiny-ImageNet datasets (results on downscaled ImageNet are in Table 4). "Standard", 'PGD' and "verified" rows report the standard test error, test error under PGD attack, and verified test error, respectively.

| Dataset | Error | CNN-7+BN | | DenseNet | | WideResNet | | ResNeXt | | Literature results | | |
|---|---|---|---|---|---|---|---|---|---|---|---|---|
| | | IBP | Ours | IBP | Ours | IBP | Ours | IBP | Ours | CROWN-IBP [57] | IBP [57][a] | Balunovic & Vechev [3] |
| CIFAR-10 $\epsilon = \frac{8}{255}$ | Standard | 57.95% | **53.71%** | 57.21% | 56.03% | 58.07% | 53.89% | 56.32% | 53.85% | 54.02% | 58.43% | 48.3% |
| | PGD | 67.10% | **64.31%** | 67.75% | 65.09% | 67.23% | 64.42% | 67.55% | 64.16% | 65.42% | 68.73% | - |
| | Verified | 69.56% | **66.62%** | 69.59% | 67.57% | 70.04% | 67.77% | 70.41% | 68.25% | 66.94% | 70.81% | 72.5% |
| Tiny-ImageNet $\epsilon = \frac{1}{255}$ | Standard | 78.54% | 78.42% | 78.40% | 77.96% | 73.54% | **72.18%** | 78.94% | 78.58% | None. [12] reported a IBP model trained on $64 \times 64$ downscaled Imagenet dataset with 84.04% clean error and 93.87% verified error. | | |
| | PGD | 81.05% | 80.96% | 80.32% | 80.52% | 79.40% | **79.48%** | 80.17% | 79.80% | | | |
| | Verified | 87.96% | 87.31% | 86.87% | 85.44% | 85.15% | **84.14%** | 87.70% | 86.95% | | | |

[a] Gowal et al. [12] reported better IBP verified error (68.44%) but this result was found not easily reproducible [57, 3]

**Robust Training of Large-scale Vision Models**  Our *loss fusion* technique allows us to scale to Tiny-ImageNet [26] and downscaled ImageNet [5]; to the best of our knowledge, this is the first LiRPA based certified defense on Tiny-ImageNet and downscaled ImageNet with a large number of class labels (200 and 1000, respectively). Besides, the automatic LiRPA bounds allow us to train certifiably robust models on complicated network architectures (WideResNet [53], DenseNet [17] and ResNeXt [52]) and

Table 4: Certified defense on Downscaled ImageNet dataset. We use WideResNet in this experiment.

| Dataset | Method | Standard | PGD | Verified |
|---|---|---|---|---|
| ImageNet ($64 \times 64$) $\epsilon = \frac{1}{255}$ | IBP [12] | 84.04% | 90.88% | 93.87% |
| | Ours | **83.77%** | **89.74%** | **91.27%** |

Table 3: Per-epoch training time and memory usage of 4 large models on CIFAR-10 with batch size 256, and 3 large models on Tiny-ImageNet with batch size 100. "LF"=loss fusion; "OOM"=out of memory. Numbers in parentheses are multiples of natural training time or memory usage. With loss fusion, LiRPA based bounds are only 3 to 5 times slower than natural training even on datasets with many labels. Without loss fusion (e.g., in [57]) LiRPA cannot scale to the Tiny-ImageNet dataset.

| Dataset | Training method | Wall Clock Time (s) | | | | GPU Memory Usage (GB) | | | |
|---|---|---|---|---|---|---|---|---|---|
| | | Natural | IBP | LiRPA w/o LF | LiRPA w/ LF | Natural | IBP | LiRPA w/o LF | LiRPA w/ LF |
| CIFAR-10 | CNN-7+BN | 11.89 | 22.23 (1.87×) | 56.05 (4.71×) | 33.40 (2.81×) | 4.42 | 7.06 (1.60×) | 20.52 (4.64×) | 10.34 (2.34×) |
| | DenseNet | 22.07 | 54.40 (2.46×) | OOM | 90.79 (4.11×) | 6.58 | 16.78 (2.55×) | OOM | 27.50 (4.18×) |
| | WideResNet | 19.39 | 43.65 (2.55×) | OOM | 74.78 (3.85×) | 7.18 | 13.50 (1.88×) | OOM | 21.98 (3.06×) |
| | ResNeXt | 14.78 | 32.44 (2.20×) | 132.70 (8.98×) | 55.84 (3.78×) | 4.74 | 11.34 (2.39×) | 43.68 (9.21×) | 18.58 (3.92×) |
| Tiny-ImageNet | CNN-7+BN | 56.70 | 112.09 (1.98×) | OOM | 163.29 (2.88×) | 4.22 | 7.12 (1.69×) | OOM | 10.57 (2.50×) |
| | DenseNet | 135.17 | 318.77 (2.36×) | OOM | 513.96 (3.80×) | 8.55 | 20.55 (2.4×) | OOM | 34.81 (4.07×) |
| | WideResNet | 133.11 | 407.74 (3.06×) | OOM | 635.50 (4.77×) | 10.91 | 24.05 (2.20×) | OOM | 39.08 (3.58×) |
| | ResNeXt | 92.63 | 191.34 (2.07×) | OOM | 337.83 (3.65×) | 4.31 | 7.05 (1.64×) | OOM | 11.66 (2.69×) |

Table 5: Verification and certified defense for LSTM and Transformer based NLP models. $\delta_{train}$ and $\delta$ represent the number of perturbed words during training and evaluation. For the most important setting $\delta_{train}=6$, we run training with 5 different seeds and report the mean and standard deviation. $\delta_{train}=0$ stands for natural training; $\delta=0$ stands for evaluating standard test accuracy. "IBP+Backward (alt.)" on $\delta_{train}=1$ has an alternative training schedule focusing on the small $\delta$ (see Appendix C.2).

| Model | Budget | Training Method | Verified Test Accuracy (%) | | | | | | |
|---|---|---|---|---|---|---|---|---|---|
| | | | $\delta=0$ | $\delta=1$ | $\delta=2$ | $\delta=3$ | $\delta=4$ | $\delta=5$ | $\delta=6$ |
| LSTM | $\delta_{train}=0$ | IBP | 84.9 | 0.6 | 0.6 | 0.6 | 0.6 | 0.6 | 0.6 |
| | | Forward | 84.9 | 0 | 0 | 0 | 0 | 0 | 0 |
| | | Forward+Backward | 84.9 | 0 | 0 | 0 | 0 | 0 | 0 |
| | $\delta_{train}=1$ | IBP | 81.3 | 78.2 | 78.2 | 78.2 | 78.2 | 78.2 | 78.2 |
| | | IBP+Backward (alt.) | 81.7 | 77.3 | 75.2 | 73.8 | 72.7 | 72.3 | 72.0 |
| | | IBP+Backward | 81.3 | 79.0 | 78.6 | 78.6 | 78.6 | 78.6 | 78.6 |
| | $\delta_{train}=6$ | IBP | 79.8±1.09 | 76.2±1.67 | 76.2±1.67 | 76.2±1.67 | 76.2±1.67 | 76.2±1.67 | 76.2±1.67 |
| | | IBP+Backward | 79.4±1.47 | 76.6±1.42 | 76.6±1.42 | 76.6±1.42 | 76.6±1.42 | 76.6±1.42 | 76.6±1.42 |
| Transformer | $\delta_{train}=0$ | IBP | 82.0 | 0.6 | 0.6 | 0.6 | 0.6 | 0.6 | 0.6 |
| | | Forward | 82.0 | 60.6 | 47.1 | 40.5 | 36.8 | 35.6 | 35.0 |
| | | Forward+Backward | 82.0 | 65.0 | 51.2 | 44.5 | 41.3 | 39.2 | 38.7 |
| | $\delta_{train}=1$ | IBP | 78.7 | 76.9 | 76.9 | 76.9 | 76.9 | 76.9 | 76.9 |
| | | IBP+Backward (alt.) | 79.2 | 77.0 | 75.4 | 75.1 | 74.5 | 74.1 | 73.9 |
| | | IBP+Backward | 78.5 | 77.3 | 77.2 | 77.1 | 77.1 | 77.1 | 77.1 |
| | $\delta_{train}=6$ | IBP | 78.4±0.34 | 76.6±0.30 | 76.6±0.30 | 76.6±0.30 | 76.6±0.30 | 76.6±0.30 | 76.6±0.30 |
| | | IBP+Backward | 78.5±0.08 | **77.4±0.21** | **77.4±0.19** | **77.4±0.19** | **77.4±0.20** | **77.4±0.20** | **77.4±0.19** |

achieve state-of-the-art results, where previous works use simpler models [50, 31, 45, 57] due to implementation difficulty. We extend CROWN-IBP [57] to the general IBP+Backward approach: we use IBP to compute bounds of intermediate nodes of the graph and use tight backward mode LiRPA for the bounds of the last layer. Unlike in CROWN-IBP, we apply loss fusion to avoid the time complexity dependency on the number of class labels, and we train a few state-of-the-art classification models ([57] used a simple CNN feedforward network). We compare our results to IBP training [12]. We provide detailed hyperparameters in Appendix C.1. We report results on CIFAR-10 [25] with $\ell_\infty$ perturbation $\epsilon = 8/255$ and Tiny-ImageNet with $\epsilon = 1/255$ in Table 2, and Downscaled-ImageNet [5] which has $1,000$ class labels with $\ell_\infty$ perturbation $\epsilon = 1/255$ in Table 4. We find that in all settings, our tight LiRPA bounds improve both clean and verified errors compared to IBP. Additionally, we achieve *state-of-the-art verified error* of 66.62% on CIFAR-10 with $\epsilon = 8/255$, better than latest published works [12, 57, 3] in certified defense.

In Table 3, we report wall clock time and GPU memory usage for regular training, pure IBP training, LiRPA training on logit layer without loss fusion (same as [57]) and LiRPA training with loss fusion. We use the same batch size 256 for all settings and conduct the experiments on 4 Nvidia GTX 1080Ti GPUs. With loss fusion, LiRPA is efficient and only 3-4 times slower than natural training on both CIFAR-10 and Tiny-ImageNet. With loss fusion, we can enable LiRPA at a cost similar to IBP, allowing us to use much tighter bounds and obtain better-verified errors than IBP (Table 2). The computational cost is significantly better than [57] which is up to 10 (number of labels) times slower than natural training on CIFAR-10, and impossible to scale to Tiny-ImageNet with 200 labels or downscaled ImageNet with 1000 labels. We also report an additional comparison where we use the largest possible batch size rather than a fixed batch size in each setting in Appendix C.1.

**Verifying and Training Robust NLP Models**  Previous works were only able to implement simple algorithms such as IBP on simple (e.g. CNN and LSTM) NLP models [20, 18] for certified defense. None of them can handle complicated models like Transformer [44] or train with tighter LiRPA bounds. We show that our algorithm can train certifiably robust models for LSTM and Transfomrer sentiment classifiers on SST-2 [41]. We consider synonym-based word substitution with $\delta \leq 6$ (up

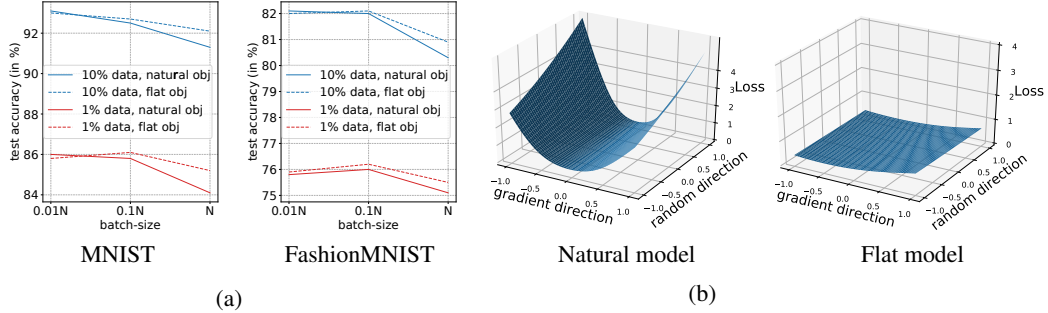

| MNIST | FashionMNIST | Natural model | Flat model |

(a) (b)

Figure 3: Application of LiRPA bounds to network parameters to obtain a model with a provably "flat" loss surface. (a) Test accuracy of naturally trained models and "flat" objective trained models on MNIST and FashionMNIST with different combinations of data size and batch size ("obj" in the legends is short for "objective"). (b) The training loss landscape of models trained with natural and flat objective on 10% data of MNIST with $0.1N$ batch size for training dataset size $N$. We plot the loss surface along the gradient direction and a random direction.

to 6 word substitutions). We provide more backgrounds and training details in Appendix C.2. In Table 5, we first verify *naturally trained* ($\delta_{\text{train}} = 0$) LSTM and Transformer. Unfortunately, most configurations cannot yield a non-trivial verified accuracy (larger than 1%), except for the case of using the forward mode and forward+backward mode perturbation analysis on a Transformer. We then conduct certified defense with $\delta_{\text{train}} = \{1, 6\}$ using IBP as in [20, 18] and our efficient IBP+Backward perturbation analysis. Models trained using IBP+Backward outperforms pure IBP (similar to our observations in computer vision tasks), and the verified test accuracy is significantly better than naturally trained models. The results demonstrate that our framework allows us to better verify and train complex NLP models using tight LiRPA bounds.

**Training Neural Networks with Guaranteed Flatness** Recently, some researchers [14, 19, 13, 16] have hypothesized that DNNs optimized with stochastic gradient descent (SGD) can find wide and flat local minima which may be associated with good generalization performance. Most previous works on LiRPA based certified defense only implemented input perturbations analysis. Our framework naturally extends to perturbation analysis on network parameters $\theta$ as they are also independent nodes in a computational graph (e.g., node $\mathbf{x}_2$ in Figure 1). This requires to relax the multiplication operation (e.g., the MatMul nodes in Figure 1) which was first discussed in Shi et al. [37], and our Algorithm 2 can then be directly applied. With this advantage, LiRPA can compute provable upper and lower bounds on the local "flatness" around a certain point $\theta_0$ for some loss $\mathcal{L}$:

$$\mathcal{L}(\theta_0) - C_L(\theta_0) \leq \mathcal{L}(\theta_0 + \Delta\theta) \leq \mathcal{L}(\theta_0) + C_U(\theta_0), \text{ for all } \|\Delta\theta\|_2 \leq \epsilon, \qquad (9)$$

where $C_L$ and $C_U$ are linear functions of $\theta_0$ that can be found by LiRPA. This is a "zeroth-order" flatness criterion, where we guarantee that the loss value does not change too much in a small region around $\theta_0$, and we do not have further assumptions on gradients or Hessian of the loss. When $\theta_0$ is a good solution, $\mathcal{L}(\theta_0)$ is close to 0, so we can simply set the left hand side of (9) to 0 and upper bound $\mathcal{L}(\theta_0 + \Delta\theta)$ to ensure flatness. Using our framework, we can train a classifier that guarantees flatness of local optimization landscape, by minimizing the "flat" objective $\mathcal{L}(\theta_0) + C_U(\theta_0)$ for the perturbation set $\mathbb{S}(\theta_0) = \{\theta : \|\theta - \theta_0\|_2 \leq \epsilon\}$ where $\theta_0$ is the current network parameter. When this "flat" objective is close to 0, we guarantee that $\mathcal{L}$ is close to 0 for all $\theta \in \mathbb{S}(\theta_0)$. We build a 3-layer MLP model with $[64, 64, 10]$ neurons in each layer and conduct experiments using only 10% and 1% of the training data in MNIST and FashionMNIST, and we then test on the full test set to aggressively evaluate the generalization performance. We also aggressively set the batch size to $\{0.01N, 0.1N, N\}$ as in [19] where $N$ is the size of training dataset. Additional details are in Appendix C.3.

The test accuracies of the models trained with regular cross entropy and our "flat" objective are shown in Figure 3a. We visualize their loss surfaces in Figure 3b. When batch size is increased or fewer data are used, test accuracy generally decreases due to overfitting, which is consistent with [22]. For models trained with the flat objective, the accuracy tends to be better, especially when a very large batch size is used. These observations provide some evidence for the hypothesis that a flat local minimum generalizes better, however, we cannot exclude the possibility that the improvements come from side effects of our objective. Our focus is to demonstrate potential applications beyond neural network verification of our framework rather than proving this hypothesis.

## Broader Impact

In this paper, we develop an automatic framework to enable perturbation analysis on any neural network structure. Our framework can be used in a wide variety of tasks ranging from robustness verification to certified defense, and potentially many more applications requiring a provable perturbation analysis. It can also play an important building block for several safety-critical ML applications, such as transportation, engineering, and healthcare, etc. We expect that our framework will significantly improve the robustness and reliability of real-world ML systems with theoretical guarantees.

An important product of this paper is an open-source LiRPA library with over 10,000 lines of code, which provides automatic and differentiable perturbation analysis. This library can tremendously facilitate the use of LiRPA for the research community as well as industrial applications, such as verifiable plant control [51]. Our library of LiRPA on general computational graphs can also inspire further improved implementations on automatic outer bounds calculations with provable guarantees.

Although our focus on this paper has been on exploring known perturbations and providing guarantees in such clairvoyant scenarios, in real-world an adversary (or nature) may not adhere to our assumptions. Thus, we may additionally want to understand implication of these unknown scenarios on the system performance. This is a relatively unexplored area in robust machine learning, and we encourage researchers to understand and mitigate the risks arising from unknown perturbations in these contexts.

## Acknowledgments and Disclosure of Funding

This work was performed under the auspices of the U.S. Department of Energy by Lawrence Livermore National Laboratory under Contract DE-AC52-07NA27344 and was partly supported by the National Science Foundation CNS-1932351, NSFC key project No. 61936010, NSFC regular project No. 61876096, NSF IIS-1901527, NSF IIS-2008173, and ARL-0011469453.

## Footnotes

[1]Note that the oracle functions of some operations also require $\underline{\mathbf{h}}_j, \overline{\mathbf{h}}_j (j \in u(i))$ for linear relaxation, although we do not explicitly mention them in the algorithm description for simplicity.

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
