[Supplementary Material]

In Appendix A, we provide more discussions on LiRPA bounds, including detailed algorithm and complexity analysis, comparison of different LiRPA implementations, and also a small numerical example in Appendix A.4. In Appendix B, we provide proofs of the theorems. We provide additional experiments, including more LiRPA trained TinyImageNet models and IBP baselines in Appendix C.1, and we also provide details for each experiment in Appendix C.

# A    Additional Discussions on LiRPA Bounds

## A.1    Oracle Functions and the Linear Relaxation of Nonlinear Operations

In this section, we summarize some examples of oracle functions as derived in previous works [54, 46, 37]. In Table 6, we provide a list of oracle functions of three basic operation types, including affine transformation, unary nonlinear function, and binary nonlinear function. Most common operations involved in neural networks can be addressed following these basic operation types. For example, dense layers and convolutional layers are affine transformations, activation functions are unary nonlinear functions, multiplication and division are binary nonlinear functions, and matrix multiplication or dot product of two variable matrices can be considered as multiplications with an affine transformation.

Parameters $\underline{\alpha}, \underline{\beta}, \underline{\gamma}, \overline{\alpha}, \overline{\beta}, \overline{\gamma}$ in Table 6 are involved in the linear relaxation of nonlinear operations. For example, for ReLU, $\sigma(h_j(\mathbf{X})) = \max(h_j(\mathbf{X}), 0)$, is a piecewise linear function and can be linearly relaxed w.r.t. the concrete bounds of $h_j(\mathbf{X})$, denoted as $l \leq h_j(\mathbf{X}) \leq u$. We aim to find parameters $\underline{\alpha}, \underline{\beta}, \overline{\alpha}, \overline{\beta}$ such that $\underline{\alpha} h_j(\mathbf{X}) + \underline{\beta} \leq \sigma(h_j(\mathbf{X})) \leq \overline{\alpha} h_j(\mathbf{X}) + \overline{\beta}$ $(\forall h_j(\mathbf{X}) \in [l, u])$ holds true. When $u \leq 0$ or $l \geq 0$, $\sigma(h_j(\mathbf{X}))$ is a linear function on $h_j(\mathbf{X}) \in [l, u]$, and $\sigma(h_j(\mathbf{X}))$ itself is the trivial linear relaxation. For $l > 0$, we have $\sigma(h_j(\mathbf{X})) = h_j(\mathbf{X})$, and thus we can take $\underline{\alpha} = \overline{\alpha} = 1, \underline{\beta} = \overline{\beta} = 0$; for $u < 0$, we have $\sigma(h_j(\mathbf{X})) \equiv 0$, and thus we can take $\underline{\alpha} = \overline{\alpha} = \underline{\beta} = \overline{\beta} = 0$. Otherwise, for $l < 0 < u$, we can take the line passing $(l, \sigma(l))$ and $(u, \sigma(u))$ as the linear upper bound, i.e., $\overline{\alpha} = \frac{\sigma(u) - \sigma(l)}{u - l}, \overline{\beta} = -\overline{\alpha} l$. For the lower bound, it can be any line with $0 \leq \underline{\alpha} \leq 1$ and $\underline{\beta} = 0$. To minimize the relaxation error, Zhang et al. [54] proposed to adaptively choose $\underline{\alpha} = I(u > |l|)$ in LiRPA. Alternatively, we can also select $\underline{\alpha} = 0$, and thereby the linear relaxation can be provably tighter than IBP bounds. This lower bound can be used for training ReLU networks with loss fusion. Figure 4 compares the linear bounds in LiRPA and IBP respesctively.

Figure 4: An example of ReLU relaxation when $l = -1.5$, $u = 1.5$. Here we take the blue dashed lines as the linear bounds, where any line passing $(0, 0)$ with a slope between 0 and 1 can be a valid lower bound. In contrast, IBP takes the fixed red dashed lines as the lower and upper bounds respectively, which is a looser relaxation.

The detailed derivation of the oracle functions shown in Table 6 has been covered in previous works [54, 46, 37] and is not a focus of this paper. We refer readers to those existing works for details.

Table 6: A list of common types of operations in neural networks, their definition $H_i$, and their corresponding oracle functions $F_i$ and $G_i$. Subscript "$_+$" stands for taking positive elements from the matrix or vector while setting other elements to zero, and vice versa for subscript "$_-$". $\text{diag}(\cdot)$ stands for constructing a diagonal matrix from a vector. $\underline{\alpha}, \underline{\beta}, \underline{\gamma}, \overline{\alpha}, \overline{\beta}, \overline{\gamma}$ are parameters of linear relaxation that can be derived for each specific nonlinear function.

| Operation Type | | Functions |
|---|---|---|
| | $H_i$ | $h_i(\mathbf{X}) = \hat{\mathbf{W}}_i h_j(\mathbf{X}) + \hat{\mathbf{b}}_i$ |
| Affine Transformation | $F_i$ | $\underline{\mathbf{\Lambda}}_j = \underline{\mathbf{A}}_i \hat{\mathbf{W}}_i$ <br> $\overline{\mathbf{\Lambda}}_j = \overline{\mathbf{A}}_i \hat{\mathbf{W}}_i$ <br> $\underline{\mathbf{\Delta}} = \underline{\mathbf{A}}_i \hat{\mathbf{b}}_i$ <br> $\overline{\mathbf{\Delta}} = \overline{\mathbf{A}}_i \hat{\mathbf{b}}_i$ |
| | $G_i$ | $\underline{\mathbf{W}}_i = \hat{\mathbf{W}}_{i,+}\underline{\mathbf{W}}_j + \hat{\mathbf{W}}_{i,-}\overline{\mathbf{W}}_j$ <br> $\underline{\mathbf{b}}_i = \hat{\mathbf{W}}_{i,+}\underline{\mathbf{b}}_j + \hat{\mathbf{W}}_{i,-}\overline{\mathbf{b}}_j + \hat{\mathbf{b}}_i$ <br> $\overline{\mathbf{W}}_i = \hat{\mathbf{W}}_{i,+}\overline{\mathbf{W}}_j + \hat{\mathbf{W}}_{i,-}\underline{\mathbf{W}}_j$ <br> $\overline{\mathbf{b}}_i = \hat{\mathbf{W}}_{i,+}\overline{\mathbf{b}}_j + \hat{\mathbf{W}}_{i,-}\underline{\mathbf{b}}_j + \hat{\mathbf{b}}_i$ |
| | $H_i$ | $h_i(\mathbf{X}) = \sigma(h_j(\mathbf{X}))$ |
| Unary Nonlinear Function | $F_i$ | $\underline{\mathbf{\Lambda}}_j = \underline{\mathbf{A}}_{i,+}\text{diag}(\underline{\alpha}) + \underline{\mathbf{A}}_{i,-}\text{diag}(\overline{\alpha})$ <br> $\overline{\mathbf{\Lambda}}_j = \overline{\mathbf{A}}_{i,+}\text{diag}(\overline{\alpha}) + \overline{\mathbf{A}}_{i,-}\text{diag}(\underline{\alpha})$ <br> $\underline{\mathbf{\Delta}} = \underline{\mathbf{A}}_{i,+}\underline{\beta} + \underline{\mathbf{A}}_{i,-}\overline{\beta}$ <br> $\overline{\mathbf{\Delta}} = \overline{\mathbf{A}}_{i,+}\overline{\beta} + \overline{\mathbf{A}}_{i,-}\underline{\beta}$ |
| | $G_i$ | $\underline{\mathbf{W}}_i = \text{diag}_+(\underline{\alpha})\underline{\mathbf{W}}_j + \text{diag}_-(\underline{\alpha})\overline{\mathbf{W}}_j$ <br> $\underline{\mathbf{b}}_i = \text{diag}_+(\underline{\alpha})\underline{\mathbf{b}}_j + \text{diag}_-(\underline{\alpha})\overline{\mathbf{b}}_j + \underline{\beta}$ <br> $\overline{\mathbf{W}}_i = \text{diag}_+(\overline{\alpha})\overline{\mathbf{W}}_j + \text{diag}_-(\overline{\alpha})\underline{\mathbf{W}}_j$ <br> $\overline{\mathbf{b}}_i = \text{diag}_+(\overline{\alpha})\overline{\mathbf{b}}_j + \text{diag}_-(\overline{\alpha})\underline{\mathbf{b}}_j + \overline{\beta}$ |
| | where | $\underline{\alpha}h_j(\mathbf{X}) + \underline{\beta} \le h_i(\mathbf{X}) \le \overline{\alpha}h_j(\mathbf{X}) + \overline{\beta}$ |
| | $H_i$ | $h_i(\mathbf{X}) = \pi(h_j(\mathbf{X}), h_k(\mathbf{X}))$ |
| Binary Nonlinear Function | $F_i$ | $\underline{\mathbf{\Lambda}}_j = \underline{\mathbf{A}}_{i,+}\text{diag}(\underline{\alpha}) + \underline{\mathbf{A}}_{i,-}\text{diag}(\overline{\alpha})$ <br> $\overline{\mathbf{\Lambda}}_j = \overline{\mathbf{A}}_{i,+}\text{diag}(\overline{\alpha}) + \overline{\mathbf{A}}_{i,-}\text{diag}(\underline{\alpha})$ <br> $\underline{\mathbf{\Lambda}}_k = \underline{\mathbf{A}}_{i,+}\text{diag}(\underline{\beta}) + \underline{\mathbf{A}}_{i,-}\text{diag}(\overline{\beta})$ <br> $\overline{\mathbf{\Lambda}}_k = \overline{\mathbf{A}}_{i,+}\text{diag}(\overline{\beta}) + \overline{\mathbf{A}}_{i,-}\text{diag}(\underline{\beta})$ <br> $\underline{\mathbf{\Delta}} = \underline{\mathbf{A}}_{i,+}\underline{\gamma} + \underline{\mathbf{A}}_{i,-}\overline{\gamma}$ <br> $\overline{\mathbf{\Delta}} = \overline{\mathbf{A}}_{i,+}\overline{\gamma} + \overline{\mathbf{A}}_{i,-}\underline{\gamma}$ |
| | $G_i$ | $\underline{\mathbf{W}}_i = \text{diag}_+(\underline{\alpha})\underline{\mathbf{W}}_j + \text{diag}_-(\underline{\alpha})\overline{\mathbf{W}}_j + \text{diag}_+(\underline{\beta})\underline{\mathbf{W}}_k + \text{diag}_-(\underline{\beta})\overline{\mathbf{W}}_k$ <br> $\underline{\mathbf{b}}_i = \text{diag}_+(\underline{\alpha})\underline{\mathbf{b}}_j + \text{diag}_-(\underline{\alpha})\overline{\mathbf{b}}_j + \underline{\beta} + \text{diag}_+(\underline{\beta})\underline{\mathbf{b}}_k + \text{diag}_-(\underline{\beta})\overline{\mathbf{b}}_k + \underline{\gamma}$ <br> $\overline{\mathbf{W}}_i = \text{diag}_+(\overline{\alpha})\overline{\mathbf{W}}_j + \text{diag}_-(\overline{\alpha})\underline{\mathbf{W}}_j + \text{diag}_+(\overline{\beta})\overline{\mathbf{W}}_k + \text{diag}_-(\overline{\beta})\underline{\mathbf{W}}_k$ <br> $\overline{\mathbf{b}}_i = \text{diag}_+(\overline{\alpha})\overline{\mathbf{b}}_j + \text{diag}_-(\overline{\alpha})\underline{\mathbf{b}}_j + \overline{\beta} + \text{diag}_+(\overline{\beta})\overline{\mathbf{b}}_k + \text{diag}_-(\overline{\beta})\underline{\mathbf{b}}_k + \overline{\gamma}$ |
| | where | $\underline{\alpha}h_j(\mathbf{X}) + \underline{\beta}h_k(\mathbf{X}) + \underline{\gamma} \le h_i(\mathbf{X}) \le \overline{\alpha}h_j(\mathbf{X}) + \overline{\beta}h_k(\mathbf{X}) + \overline{\gamma}$ |

## A.2 Complexity Comparison between Different Perturbation Analysis Modes

In this section, we compare the computational cost of different perturbation analysis modes. We assume that $D_x$ and $D_y$ are the total dimension of the perturbed independent nodes and the final output node respectively. We focus on a usual case in classification models, where the final output node is a logits layer whose dimension equals to the number of classes and thus usually $D_y \ll D_x$ holds true, or the final output is a loss function with $D_y = 1 \ll D_x$ if loss fusion is enabled. We also assume that the time complexity of a regular forward pass of the computational graph (e.g., a regular inference pass) is $O(r)$, and the complexity of a regular back propagation pass in gradient computation is also asymptotically $O(r)$. Note that the overall time complexity of LiRPA depends on oracle functions, and in the below analysis we focus on common cases (e.g., common activation functions in Table 6).

**Interval bound propagation (IBP)**   IBP can be seen as a special and degenerated case of LiRPA bounds. The time complexity of pure IBP is still $O(r)$ since it computes two output values, a lower bound and a upper bound, for each neuron, and thus the time complexity is the same as a regular forward pass which computes one output value for each neuron. However, pure IBP cannot give tight enough bounds for models without certifiably robust training or during the early stage of robust training.

**Backward mode bound propagation** Backward mode LiRPA oracles typically require bounds of intermediate nodes $\underline{\mathbf{h}}_j$, $\overline{\mathbf{h}}_j$ for all $j \in u(i)$ for a node $i$ (referred to as "pre-activation bounds" in some works). In the *IBP+Backward* setting, we assume that the intermediate bounds are known from IBP before using the backward mode LiRPA. The oracle function $F_i$ typically has the same time complexity as back propagation of gradients through node $i$ (e.g., for linear layers it is the transposed operation of $H_i(\cdot)$). However, unlike in back propagation where the gradients is computed for a scalar function, in backward mode LiRPA we need to compute $O(D_y)$ values for each neuron, and these values stand for the coefficients of the linear bounds of the $D_y$ final output neurons. The time complexity is roughly $D_y$ times back propagation time, $O(D_y r)$.

For a purely backward perturbation analysis that can be extended from CROWN [54], the bounds of intermediate nodes needed for the oracle functions are also computed with a backward mode LiRPA. Assuming there are $N$ nodes in total (including output nodes and all intermediate nodes) that require LiRPA bounds, the total time complexity is asymptotically $O(Nr)$ where $N$ can be a quite large number (e.g., for feed-forward ReLU networks $N$ includes hidden neurons over all layers and $N \gg D_y$), so this approach cannot scale to large graphs or be used for efficient training.

**Forward mode bound propagation** In the forward mode perturbation analysis, since we represent the bounds of each neuron with linear functions w.r.t. the perturbed independent nodes, we need to compute $O(D_x)$ values for each neuron. Usually, the oracle functions $G_i$ has the same asymptotic complexity as the computation function $H_i(\cdot)$; however, the inputs of $G_i$ include dimension $D_x$, and the total time complexity of is roughly $O(D_x r)$. Note that in the implementation of the forward mode, we do not compute linear functions w.r.t. all the independent nodes, but we only need to consider those perturbed independent nodes while treating the other independent nodes as constants, and thereby $D_x$ may be much smaller than the dimension of $\mathbf{X}$, e.g., model parameters can be excluded if they are not perturbed.

**Efficient hybrid bounds** Among the LiRPA variants, *IBP+Backward* with a complexity of $O(D_y r)$ is usually most efficient for classification models and is used in our certified training experiments. When loss fusion is enabled, $D_y = 1$ during training, and thereby the complexity of *IBP+Backward* is $O(r)$, which is the same as that of IBP. In this way, our loss fusion technique can significantly improve the scalability of certified training with LiRPA bounds. To obtain tighter bounds for intermediate nodes which can also tighten the final output bounds, we may use pure forward or *Forward+Backward* mode with a complexity of $O((D_x + D_y)r)$ which is usually larger than that of *IBP+Backward* when $D_y \ll D_x$. The forward mode LiRPA can still be potentially useful for situations where $D_x \ll D_y$, e.g., for generative models with a large output dimension. We leave this as future work.

## A.3  The GetOutDegree Auxiliary Function in Backward Mode Perturbation Analysis

---
**Algorithm 3** Auxiliary Function for Computing Output Degrees

---
   **function** GetOutDegree ($o$)
      Create BFS queue and $Q.push(o)$
      $d_i \leftarrow 0 \ (\forall i \leq n)$
      **while** $Q$ is not empty **do**
         $i = Q.pop()$
         **for** $j \in u(i)$ **do**
            $d_j \mathrel{+}= 1$
            **if** $j$ has not been in $Q$ **then**
               $Q.push(j)$

---

As mentioned in Section 3.4, we have an auxiliary "GetOutDegree" function for computing the degree $d_i$ of each node $i$, which is defined as the the number of outputs nodes of node $i$ that the node $o$ is dependent on. This function is illustrated in Algorithm 3. We use a BFS pass. At the beginning, node $o$ is added into the queue. Next, each time we pick a node $i$ from the head of the queue. Node $o$ is dependent on node $i$, and thus we increase the degree of its input nodes, each $d_j (j \in u(i))$, by 1. Node $o$ is also dependent on node $j (j \in u(i))$ and we add node $j$ to the queue if it has never been in the queue yet. We repeat this process until the queue becomes empty, and at this time any node $i$ that node $o$ is dependent on has been visited and has contributed to the $d_j (j \in u(i))$ of its input nodes.

### A.4 A Small Example of LiRPA Bounds

We provide a small example to illustrate the computation of our LiRPA methods. We assume that we have a simple ReLU network with 2 hidden layers, with weight matrix of each layer as below:

$$\hat{\mathbf{W}}_1 = [[2,1],[-3,4]], \ \ \hat{\mathbf{W}}_2 = [[4,-2],[2,1]], \ \ \hat{\mathbf{W}}_3 = [-2,1],$$

and we do not consider bias terms of the layers here for simplicity.

Given a clean input $\mathbf{X}_0 = [[0],[1]]$ and $\ell_\infty$ perturbation with $\epsilon = 2$, we can compute the bounds of the last layer and compare the results from IBP, forward mode LiRPA and backward mode LiRPA respectively.

**IBP**

$$\overline{\mathbf{h}}_1 = [[2],[3]],$$
$$\underline{\mathbf{h}}_1 = [[-2],[-1]],$$
$$\overline{\mathbf{h}}_2 = \hat{\mathbf{W}}_{1,+}\overline{\mathbf{h}}_1 + \hat{\mathbf{W}}_{1,-}\underline{\mathbf{h}}_1 = [[7],[12]] + [[0],[6]] = [[7],[18]],$$
$$\underline{\mathbf{h}}_2 = \hat{\mathbf{W}}_{1,+}\underline{\mathbf{h}}_1 + \hat{\mathbf{W}}_{1,-}\overline{\mathbf{h}}_1 = [[-5],[-4]] + [[0],[-6]] = [[-5],[-10]],$$
$$\overline{\mathbf{h}}_3 = \hat{\mathbf{W}}_{2,+}\overline{\mathbf{h}}_2 + \hat{\mathbf{W}}_{2,-}\underline{\mathbf{h}}_2 = [[28],[32]] + [[0],[0]] = [[28],[32]],$$
$$\underline{\mathbf{h}}_3 = \hat{\mathbf{W}}_{2,+}\underline{\mathbf{h}}_2 + \hat{\mathbf{W}}_{2,-}\overline{\mathbf{h}}_2 = [[0],[0]] + [[-36],[0]] = [[-36],[0]],$$
$$\overline{\mathbf{h}}_4 = \hat{\mathbf{W}}_{3,+}\overline{\mathbf{h}}_3 + \hat{\mathbf{W}}_{3,-}\underline{\mathbf{h}}_3 = [32] + [0] = [32],$$
$$\underline{\mathbf{h}}_4 = \hat{\mathbf{W}}_{3,+}\underline{\mathbf{h}}_3 + \hat{\mathbf{W}}_{3,-}\overline{\mathbf{h}}_3 = [0] + [-56] = [-56].$$

In the following computation of LiRPA bounds, we always use $\underline{\alpha} = 0$ in the linear relaxation of ReLU activation.

**Forward Mode LiRPA**

$$\overline{\mathbf{W}}_1 = \underline{\mathbf{W}}_1 = \mathbf{I}, \ \ \underline{\mathbf{b}}_1 = \overline{\mathbf{b}}_1 = \mathbf{0},$$
$$\overline{\mathbf{W}}_2 = \underline{\mathbf{W}}_2 = \hat{\mathbf{W}}_1 = [[2,1],[-3,4]],$$
$$\overline{\mathbf{h}}_2 = 2[[3],[7]] + [[1],[4]] = [[7],[18]],$$
$$\underline{\mathbf{h}}_2 = -2[[3],[7]] + [[1],[4]] = [[-5],[-10]].$$

We compute the relaxation of the first layer ReLU activations:

$$\text{diag}(\overline{\alpha}_1) = [[0.58,0],[0,0.64]], \ \ \text{diag}(\underline{\alpha}_1) = [[0,0],[0,0]],$$
$$\overline{\beta_1} = [[2.92],[6.43]]], \ \ \underline{\beta_1} = [[0],[0]],$$

and then we have:

$$\overline{\mathbf{W}}_3 = \hat{\mathbf{W}}_{2,+}(\text{diag}(\overline{\alpha}_1)\overline{\mathbf{W}}_2) + \hat{\mathbf{W}}_{2,-}(\text{diag}(\underline{\alpha}_1)\underline{\mathbf{W}}_2) = [[4.67,2.33],[0.40,3.74]],$$
$$\underline{\mathbf{W}}_3 = \hat{\mathbf{W}}_{2,-}(\text{diag}(\overline{\alpha}_1)\overline{\mathbf{W}}_2) + \hat{\mathbf{W}}_{2,+}(\text{diag}(\underline{\alpha}_1)\underline{\mathbf{W}}_2) = [[3.86,-5.14],[0,0]],$$
$$\overline{\mathbf{d}}_2 = \hat{\mathbf{W}}_{2,+}\overline{\beta}_1 + \hat{\mathbf{W}}_{2,-}\underline{\beta}_1 = [[11.67],[12.26]],$$
$$\underline{\mathbf{d}}_2 = \hat{\mathbf{W}}_{2,-}\overline{\beta}_1 + \hat{\mathbf{W}}_{2,+}\underline{\beta}_1 = [[-12.86],[0]],$$
$$\overline{\mathbf{h}}_3 = \overline{\mathbf{W}}_3\mathbf{X}_0 + \|\overline{\mathbf{W}}_3\|_1\epsilon + \overline{\mathbf{d}}_2 = [[28],[24]],$$
$$\underline{\mathbf{h}}_3 = \underline{\mathbf{W}}_3\mathbf{X}_0 - \|\underline{\mathbf{W}}_3\|_1\epsilon + \underline{\mathbf{d}}_2 = [[-36],[0]].$$

We then repeat the computation on the second layer:

$$\text{diag}(\overline{\alpha}_2) = [[0.4375,0],[0,1]], \ \ \text{diag}(\underline{\alpha}_2) = [[0,0],[0,1],]$$
$$\overline{\beta_2} = [[15.75],[0]], \ \ \underline{\beta_2} = [[0],[0]],$$

$$\overline{\mathbf{W}}_4 = \hat{\mathbf{W}}_{3,+}(\text{diag}(\overline{\alpha}_2)\overline{\mathbf{W}}_3) + \hat{\mathbf{W}}_{3,-}(\text{diag}(\underline{\alpha}_2)\underline{\mathbf{W}}_3) = [0.40, 3.74],$$

$$\underline{\mathbf{W}}_4 = \hat{\mathbf{W}}_{3,-}(\text{diag}(\overline{\alpha}_2)\overline{\mathbf{W}}_3) + \hat{\mathbf{W}}_{3,+}(\text{diag}(\underline{\alpha}_2)\underline{\mathbf{W}}_3) = [-4.08, -2.04],$$

$$\overline{\mathbf{d}}_3 = \hat{\mathbf{W}}_{3,+}(\overline{\beta}_2 + \text{diag}(\overline{\alpha}_2)\overline{\beta}_2) + \hat{\mathbf{W}}_{3,-}(\underline{\beta}_2 + \text{diag}(\underline{\alpha}_2)\underline{\beta}_2) = [12.26],$$

$$\underline{\mathbf{d}}_3 = \hat{\mathbf{W}}_{3,-}(\overline{\beta}_2 + \text{diag}(\overline{\alpha}_2)\overline{\beta}_2) + \hat{\mathbf{W}}_{3,+}(\underline{\beta}_2 + \text{diag}(\underline{\alpha}_2)\underline{\beta}_2) = [-41.71],$$

$$\overline{\mathbf{h}}_4 = \overline{\mathbf{W}}_4\mathbf{X}_0 + \|\overline{\mathbf{W}}_4\|_1\epsilon + \overline{\mathbf{d}}_3 = [24.29],$$

$$\underline{\mathbf{h}}_4 = \underline{\mathbf{W}}_4\mathbf{X}_0 - \|\underline{\mathbf{W}}_4\|_1\epsilon + \underline{\mathbf{d}}_3 = [-56].$$

**Backward Mode LiRPA**  Here we reuse the intermediate results from the forward mode LiRPA for the linear relaxation of ReLU activations, where

$$\text{diag}(\overline{\alpha}_1) = [[0.58, 0], [0, 0.64]], \quad \text{diag}(\underline{\alpha}_1) = [[0, 0], [0, 0]],$$

$$\overline{\beta}_1 = [[2.92], [6.43]]], \quad \underline{\beta}_1 = [[0], [0]],$$

$$\text{diag}(\overline{\alpha}_2) = [[0.4375, 0], [0, 1]], \quad \text{diag}(\underline{\alpha}_2) = [[0, 0], [0, 1]]$$

$$\overline{\beta}_2 = [[15.75], [0]], \quad \underline{\beta}_2 = [[0], [0]].$$

We then compute the linear bounds from the last layer to the first layer and finally concretize the linear bounds:

$$\underline{\mathbf{A}}_4 = \overline{\mathbf{A}}_4 = \mathbf{I},$$

$$\underline{\mathbf{A}}_3 = \underline{\mathbf{A}}_4\hat{\mathbf{W}}_3 = [-2, 1],$$

$$\overline{\mathbf{A}}_3 = \overline{\mathbf{A}}_4\hat{\mathbf{W}}_3 = [-2, 1],$$

$$\overline{\mathbf{A}}_2 = \overline{\mathbf{A}}_{3,+}\text{diag}(\overline{\alpha}_2)\hat{\mathbf{W}}_2 + \overline{\mathbf{A}}_{3,-}\text{diag}(\underline{\alpha}_2)\hat{\mathbf{W}}_2 = [2, 1],$$

$$\underline{\mathbf{A}}_2 = \underline{\mathbf{A}}_{3,+}\text{diag}(\underline{\alpha}_2)\hat{\mathbf{W}}_2 + \underline{\mathbf{A}}_{3,-}\text{diag}(\overline{\alpha}_2)\hat{\mathbf{W}}_2 = [-1.5, 2.75],$$

$$\overline{\mathbf{A}}_1 = \overline{\mathbf{A}}_{2,+}\text{diag}(\overline{\alpha}_1)\hat{\mathbf{W}}_1 + \overline{\mathbf{A}}_{2,-}\text{diag}(\underline{\alpha}_1)\hat{\mathbf{W}}_1 = [0.40, 3.74],$$

$$\underline{\mathbf{A}}_1 = \underline{\mathbf{A}}_{2,+}\text{diag}(\underline{\alpha}_1)\hat{\mathbf{W}}_1 + \underline{\mathbf{A}}_{2,-}\text{diag}(\overline{\alpha}_1)\hat{\mathbf{W}}_1 = [-1.75, -0.875],$$

$$\overline{\mathbf{d}}_1 = \overline{\mathbf{A}}_{2,+}\overline{\beta}_2 + \overline{\mathbf{A}}_{2,-}\underline{\beta}_2 + \overline{\mathbf{A}}_{1,+}\overline{\beta}_1 + \overline{\mathbf{A}}_{1,-}\underline{\beta}_1 = [12.26],$$

$$\underline{\mathbf{d}}_1 = \underline{\mathbf{A}}_{2,+}\underline{\beta}_2 + \underline{\mathbf{A}}_{2,-}\overline{\beta}_2 + \underline{\mathbf{A}}_{1,+}\underline{\beta}_1 + \underline{\mathbf{A}}_{1,-}\overline{\beta}_1 = [-35.875],$$

$$\overline{\mathbf{h}}_4 = \overline{\mathbf{A}}_1\mathbf{X}_0 + \|\overline{\mathbf{A}}_1\|_1\epsilon + \overline{\mathbf{d}}_1 = [24.28],$$

$$\underline{\mathbf{h}}_4 = \underline{\mathbf{A}}_1\mathbf{X}_0 - \|\underline{\mathbf{A}}_1\|_1\epsilon + \underline{\mathbf{d}}_1 = [-42].$$

As we can see from this example, the bounds from the backward mode LiRPA are the tightest compared to those from forward mode LiRPA and IBP, even if we reuse the intermediate relaxation results from the forward mode LiRPA.

## A.5  Existing LiRPA implementations

We list and compare a few notable LiRPA implementations in Table 7.

Table 7: Comparison between different implementations for perturbation analysis. ("FF" = FeedForward network).

| Method | Based On | Mode | Structure | Activation | Perturbation | Differentiability | Automatic[a] | Efficiency | Tightness |
|---|---|---|---|---|---|---|---|---|---|
| DiffAI [31] | PyTorch | Backward, IBP | FF+ResNet | ReLU | $\ell_\infty$ | Yes | No | GPU | ++ |
| IBP [12, 31] | TensorFlow | IBP | General | General | $\ell_\infty$ | Yes | No | GPU | - |
| ERAN [30] | C++/CUDA[b] | Backward, IBP, others[c] | General | General | $\ell_p$+semantic | No | No | Partially GPU | ++ |
| Convex-Adv [49] | PyTorch | Backward | FF+ResNet | ReLU | $\ell_p$ | Yes | No | Multi-GPU | + |
| Fast-Lin [47] | Numpy | Backward | FF (MLP) | ReLU | $\ell_p$ | No | No | CPU | + |
| CROWN [54] | Numpy | Backward | FF (MLP) | General | $\ell_p$ | No | No | CPU | ++ |
| CROWN-IBP [54] | PyTorch | Backward, IBP | FF | General | $\ell_p$ | Yes | No | Multi-GPU | ++ |
| Ours | PyTorch | Backward, Forward, IBP | General | General | General[d] | Yes | Yes | Multi-GPU | ++ |

[a] "Automatic" is defined as an user can easily obtain bounds using existing model source code, without manual conversion or implementation.
[b] ERAN has a TensorFlow frontend to read TensorFlow models, but its backend is written in C++ and partially CUDA.
[c] Other types of bounds like k-ReLU [39] are provided, but typically much less efficient than IBP or backward mode perturbation analysis.
[d] User supplied perturbation specifications.

# B Proofs of the Theorems

## B.1 Proof of Theorem 1

In Theorem 1, we bound node $o$ with:

$$\sum_{i \in \mathbf{V}} \underline{\mathbf{A}}_i h_i(\mathbf{X}) + \underline{\mathbf{d}} \le h_o(\mathbf{X}) \le \sum_{i \in \mathbf{V}} \overline{\mathbf{A}}_i h_i(\mathbf{X}) + \overline{\mathbf{d}} \quad \forall \mathbf{X} \in \mathbb{S}. \tag{10}$$

Initially, this inequality holds true with

$$\underline{\mathbf{A}}_o = \overline{\mathbf{A}}_o = \mathbf{I}, \quad \underline{\mathbf{A}}_i = \overline{\mathbf{A}}_i = \mathbf{0}(i \neq o), \quad \underline{\mathbf{d}} = \overline{\mathbf{d}} = \mathbf{0}, \tag{11}$$

because then

$$\sum_{i \in \mathbf{V}} \underline{\mathbf{A}}_i h_i(\mathbf{X}) + \underline{\mathbf{d}} = \sum_{i \in \mathbf{V}} \overline{\mathbf{A}}_i h_i(\mathbf{X}) + \overline{\mathbf{d}} = h_o(\mathbf{X})$$

meets (10).

Without loss of generality, we assume that the nodes are numbered in topological order, i.e., for each node $i$ and its input node $j \in u(i)$, $i > j$ holds true, and we assume that there are $n'$ independent nodes. Then, we have $o = n$, and all the independent nodes have the smallest numbers compared to the other nodes. This can be achieved via a topological sort for any computational graph. We can also ignore nodes that node $o$ does not depend on. With these assumptions, we show a lemma:

**Lemma 4.** *In Algorithm 2, every dependent node $i(n' < i \le n)$ will be visited once and only once. And when node $i$ is visited, all nodes that depend on node $i$ must have been visited.*

*Proof.* First, node $o$ is added to the queue and will be visited, and since it has no successor node, it will not be added to the queue again during the BFS. We assume that node $i \ldots n$ will be visited once and only once, and this is initially true with $i = o = n$. For node $i - 1 > n'$, we show that node $(i-1)$ will also be visited once and only once. When node $i \ldots n$ have all been visited, the successor nodes of node $(i-1)$ have been visited and $d_{i-1} = 0$, and node $(i-1)$ is a dependent node. Therefore, node $(i-1)$ will be added to the queue and visited. From the assumption on node $i \ldots n$, all nodes that depend on the successor nodes of node $(i-1)$ have also been visited. Nodes that depend on node $(i-1)$ consist of the successor nodes of node $(i-1)$ and nodes that depend on these successors, and thus they have all been visited. Since node $i \ldots n$ will not be visited more than once, node $(i-1)$ will not be added to the queue by its successor nodes more than once. Therefore, node $(i-1)$ will also be visited once and only once. Using mathematical induction, we can prove that the lemma holds true for all node $i(n' < i \le n')$. $\square$

According to Lemma 4, every dependent node $i$ is visited once and exactly once. When node $i$ is visited, Algorithm 2 performs the following changes to attributes $\underline{\mathbf{d}}$, $\overline{\mathbf{d}}$, $\underline{\mathbf{A}}_i$, $\overline{\mathbf{A}}_i$ and $\underline{\mathbf{A}}_j, \overline{\mathbf{A}}_j (\forall j \in u(i))$:

$$\underline{\mathbf{A}}_j \mathrel{+}= \underline{\mathbf{\Lambda}}_j, \quad \overline{\mathbf{A}}_j \mathrel{+}= \overline{\mathbf{\Lambda}}_j, \quad d_j \mathrel{-}= 1 \quad \forall j \in u(i), \tag{12}$$

$$\underline{\mathbf{d}} \mathrel{+}= \underline{\mathbf{\Delta}}, \quad \overline{\mathbf{d}} \mathrel{+}= \overline{\mathbf{\Delta}}, \quad \underline{\mathbf{A}}_i \leftarrow \mathbf{0}, \quad \overline{\mathbf{A}}_i \leftarrow \mathbf{0}, \tag{13}$$

where $\underline{\mathbf{\Lambda}}_j, \overline{\mathbf{\Lambda}}_j, \underline{\mathbf{\Delta}}_j, \overline{\mathbf{\Delta}}_j$ come from oracle function $F_i$ as shown in (5), and

$$\sum_{j \in u(i)} \underline{\mathbf{\Lambda}}_j h_j(\mathbf{X}) + \underline{\mathbf{\Delta}} \le \underline{\mathbf{A}}_i h_i(\mathbf{X}), \quad \overline{\mathbf{A}}_i h_i(\mathbf{X}) \le \sum_{j \in u(i)} \overline{\mathbf{\Lambda}}_j h_j(\mathbf{X}) + \overline{\mathbf{\Delta}}.$$

Thereby, with changes in (12) and (13), the linear lower bound in (10) becomes

$$h_o(\mathbf{X}) \ge \sum_{k \in \mathbf{V}} \underline{\mathbf{A}}_k h_k(\mathbf{X}) + \underline{\mathbf{d}}$$

$$= \sum_{k \in \mathbf{V}, k \neq i, k \notin u(i)} \underline{\mathbf{A}}_k h_k(\mathbf{X}) + \sum_{j \in u(i)} \underline{\mathbf{A}}_j h_j(\mathbf{X}) + \underline{\mathbf{A}}_i h_i(\mathbf{X}) + \underline{\mathbf{d}}$$

$$\ge \sum_{k \in \mathbf{V}, k \neq i, k \notin u(i)} \underline{\mathbf{A}}_k h_k(\mathbf{X}) + \sum_{j \in u(i)} \underline{\mathbf{A}}_j h_j(\mathbf{X}) + \sum_{j \in u(i)} \underline{\mathbf{\Lambda}}_j h_j(\mathbf{X}) + \underline{\mathbf{\Delta}} + \underline{\mathbf{d}}$$

$$= \sum_{k \in \mathbf{V}, k \neq i, k \notin u(i)} \underline{\mathbf{A}}_k h_k(\mathbf{X}) + \sum_{j \in u(i)} (\underline{\mathbf{A}}_j + \underline{\mathbf{\Lambda}}_j) h_j(\mathbf{X}) + (\underline{\mathbf{\Delta}} + \underline{\mathbf{d}}), \tag{14}$$

which remains a valid linear lower bound in the form of (10). Similarly, this also holds true for the linear upper bound. In this way, $\underline{\mathbf{A}}_i$ and $\overline{\mathbf{A}}_i$ are propagated to its input nodes and set to $\mathbf{0}$. Thereby the term w.r.t. $h_i(\mathbf{X})$ is eliminated in the linear bounds.

At this time, all successor nodes of node $i$ have been visited and will not been visited again. Therefore, $\underline{\mathbf{A}}_i$ and $\overline{\mathbf{A}}_i$ will keep to be $\mathbf{0}$ after node $i$ is visited. Therefore, when Algorithm 2 terminates, $\underline{\mathbf{A}}_i, \overline{\mathbf{A}}_i$ of all dependent node $i$ will be $\mathbf{0}$, and thereby we will obtain linear bounds of node $o$ w.r.t. all the independent nodes.

## B.2 Proof of Theorem 2

Theorem 2 shows that linear bounds under perturbation defined by synonym-based word substitution can be concretized with a dynamic programming. Specifically, to concretize a linear lower bound, we need to compute

$$\underline{\mathbf{h}}_o = \min_{\hat{w}_1, \hat{w}_2, \ldots, \hat{w}_n} \underline{\mathbf{b}}_o + \sum_{t=1}^{n} \tilde{\underline{\mathbf{W}}}_t e(\hat{w}_t) \quad \text{s.t.} \quad \sum_{t=1}^{n} I(\hat{w}_t \neq w_t) \leq \delta, \tag{15}$$

where $e(\hat{w}_t)$ is the embedding of the $t$-th word in the input, $\tilde{\underline{\mathbf{W}}}_t$ consists of columns in $\underline{\mathbf{W}}_o$ corresponding to the $e(\hat{w}_t)$ term in the linear bound. In the dynamic programming, we compute $\underline{\mathbf{g}}_{i,j}(j \leq i)$ that denotes the lower bound of $\underline{\mathbf{b}}_o + \sum_{t=1}^{i} \tilde{\underline{\mathbf{W}}}_t e(\hat{w}_t)$ when $j$ words among the first $i$ words $\hat{w}_1, \ldots, \hat{w}_i$ have been replaced. If $\hat{w}_k$ has not been replaced, $\hat{w}_k = w_k$, otherwise $\hat{w}_k \in \mathbb{S}(w_k)$. For $i = 0$, obviously $\underline{\mathbf{g}}_{0,0} = \underline{\mathbf{b}}_o$. For $j = 0$, $\hat{w}_1, \hat{w}_2, \cdots, \hat{w}_i$ must have not been replaced and thus $\hat{w}_t = w_t(1 \leq t \leq i)$ holds true. Therefore, $\underline{\mathbf{g}}_{i,0} = \underline{\mathbf{b}}_o + \sum_{t=1}^{i} \tilde{\underline{\mathbf{W}}}_t e(w_t)$. For $i, j > 0$, we consider whether $\hat{w}_i$ has been replaced. If $\hat{w}_i$ has not been replaced, $\tilde{\underline{\mathbf{W}}}_i e(\hat{w}_i) = \tilde{\underline{\mathbf{W}}}_i e(w_i)$, and $j$ words have been replaced among the first $i-1$ words. In this case, $\underline{\mathbf{b}}_o + \sum_{t=1}^{i} \tilde{\underline{\mathbf{W}}}_t e(\hat{w}_t) = \underline{\mathbf{b}}_o + \sum_{t=1}^{i-1} \tilde{\underline{\mathbf{W}}}_t e(\hat{w}_t) + \tilde{\underline{\mathbf{W}}}_i e(w_i) \geq \underline{\mathbf{g}}_{i-1,j} + \tilde{\underline{\mathbf{W}}}_i e(w_i)$. For the other case if $\hat{w}_i$ has been replaced, $j - 1$ words have been replaced among the first $i - 1$ words, and $\underline{\mathbf{b}}_o + \sum_{t=1}^{i} \tilde{\underline{\mathbf{W}}}_t e(\hat{w}_t) \geq \underline{\mathbf{g}}_{i-1,j-1} + \min_{w'}\{\tilde{\underline{\mathbf{W}}}_i e(w')\}$, where $w' \in \mathbb{S}(w_i)$. We combine these two cases and take the minimum of their results, and thus:

$$\underline{\mathbf{g}}_{i,j} = \min(\underline{\mathbf{g}}_{i-1,j} + \tilde{\underline{\mathbf{W}}}_i e(w_i), \ \underline{\mathbf{g}}_{i-1,j-1} + \min_{w'}\{\tilde{\underline{\mathbf{W}}}_i e(w')\}) \ (i, j > 0) \quad \text{s.t.} \ w' \in \mathbb{S}(w_i).$$

The result of (15) is $\min_{j=0}^{\delta} \underline{\mathbf{g}}_{n,j}$. The upper bounds can also be computed in a similar way simply by changing from taking the minimum to taking the maximum in the above derivation.

## B.3 Proof of Theorem 3

In Theorem 3, we show that given concrete lower and upper bounds of $g_\theta(\mathbf{X}, y)$ as $\underline{g}_\theta(\mathbf{X}, y)$ and $\overline{g}_\theta(\mathbf{X}, y)$, with $S(\mathbf{X}, y) = \sum_{i \leq K} \exp(-[g_\theta(\mathbf{X}, y)]_i)$, we have

$$\max_{\mathbf{X} \in \mathbb{S}} L(f_\theta(\mathbf{X}), y) \leq \log \overline{S}(\mathbf{X}, y) \leq L(-\underline{g}_\theta(\mathbf{X}, y), y), \tag{16}$$

where $\overline{S}(\mathbf{X}, y)$ is the upper bound of $S(\mathbf{X}, y)$ from the backward mode LiRPA.

$L(f_\theta(\mathbf{X}), y)$ is the cross entropy loss with softmax normalization, and

$$L(f_\theta(\mathbf{X}), y) = -\log \frac{[\exp(f_\theta(\mathbf{X}))]_y}{\sum_{i \leq K}[\exp(f_\theta(\mathbf{X}))]_i}$$

$$= \log \sum_{i \leq K} \exp([f_\theta(\mathbf{X})]_i - [f_\theta(\mathbf{X})]_y)$$

$$= \log \sum_{i \leq K} \exp(-[g_\theta(\mathbf{X}, y)]_i)$$

$$= \log S(\mathbf{X}, y).$$

Since $\log$ is a monotonic function,

$$\max_{\mathbf{X} \in \mathbb{S}} L(f_\theta(\mathbf{X}), y) = \log \max_{\mathbf{X} \in \mathbb{S}} S(\mathbf{X}, y) \leq \log \overline{S}(\mathbf{X}, y).$$

And $L(-\underline{g}_\theta(\mathbf{X}, y), y)$ is an upper bound of $\max_{\mathbf{X} \in \mathbb{S}} L(f_\theta(\mathbf{X}), y)$, since

$$\max_{\mathbf{X} \in \mathbb{S}} L(f_\theta(\mathbf{X}), y) \leq \log \sum_{i \leq K} \exp(-\min_{\mathbf{X} \in \mathbb{S}}[g_\theta(\mathbf{X}, y)]_i)$$

$$\leq \log \sum_{i \leq K} \exp(-[\underline{g}_\theta(\mathbf{X}, y)]_i)$$

$$= L(-\underline{g}_\theta(\mathbf{X}, y), y).$$

Figure 5: Illustration of different upper bounds of $\exp(x)$ within $x \in [-1.5, 1.5]$. The linear bound (blue line) is a tighter bound than the IBP bound (red line). The blue area stands for the gap between the two upper bounds. Note that for this particular setting of upper bounding $\overline{S}(\mathbf{X}, y)$ we need only upper bounds for this non-linear function.

Now we are going to show that $\log \overline{S}(\mathbf{X}, y) \leq L(-\underline{g}_\theta(\mathbf{X}, y), y)$. Here we assume that the concrete bounds of intermediate layers used for linear relaxations and also the concrete lower and upper bounds of $g_\theta(\mathbf{X}, y)$ (denoted as $\underline{g}_\theta(\mathbf{X}, y)$ and $\overline{g}_\theta(\mathbf{X}, y)$) are the same.

Computing $\sum_{i \leq K} \exp(-[\underline{g}_\theta(\mathbf{X}, y)]_i)$ is essentially propagating $\underline{g}_\theta(\mathbf{X}, y)$ through exp and summation in the loss function using IBP, while $\overline{S}(\mathbf{X}, y)$ is directly computed from the LiRPA bound of $S(\mathbf{X}, y)$. Using $\tilde{\mathbf{A}}$, a matrix of ones with size $1 \times K$, to replace the summation, we can unify these two processes as computing the upper bound of $\tilde{\mathbf{A}} \exp(-g_\theta(\mathbf{X}, y))$ using LiRPA with different relaxations for exp. For $\overline{S}(\mathbf{X}, y)$, the linear upper bound of $\exp(x)(l \leq x \leq u)$ is a line passing $(l, e^l)$ and $(u, e^u)$, while it is constant $e^u$ when computing $\sum_{i \leq K} \exp(-[\underline{g}_\theta(\mathbf{X}, y)]_i)$. We illustrate the two different relaxations in Figure 5. Since elements in $\tilde{\mathbf{A}}$ are all positive, the lower bound of $\exp(x)$ will not be involved, and thus with the same concrete bounds of $g_\theta$ the relaxation on exp in $\overline{S}(\mathbf{X}, y)$ is strictly tighter when $l < u$.

After relaxing exp, we can obtain two linear upper bounds $\hat{\mathbf{A}} g_\theta(\mathbf{X}, y) + \hat{\mathbf{d}}$ from the two methods respectively, where $\hat{\mathbf{A}}$ and $\hat{\mathbf{d}}$ are obtained by merging the relaxation of exp and $\tilde{\mathbf{A}}$. Note that since the relaxed function $\exp(x) \leq e^u$ in IBP has no linear term, in this case $\hat{\mathbf{A}} = \mathbf{0}$ and the upper bound will simply be $\hat{\mathbf{d}}$. We then back propagate $\hat{\mathbf{A}} g_\theta(\mathbf{X}, y) + \hat{\mathbf{d}}$ to the input and concretize the bounds to get $\overline{S}(\mathbf{X}, y)$ and $\sum_{i \leq K} \exp(-[\underline{g}_\theta(\mathbf{X}, y)]_i)$ respectively. In the calculation of linear bounds, the linear relaxations of all the other nonlinear operations are the same for $\overline{S}(\mathbf{X}, y)$ and $\sum_{i \leq K} \exp(-[\underline{g}_\theta(\mathbf{X}, y)]_i)$ while the exp relaxation is the only difference. Since the relaxation for $\overline{S}(\mathbf{X}, y)$ is no looser than that for $\sum_{i \leq K} \exp(-[\underline{g}_\theta(\mathbf{X}, y)]_i)$, the upper linear bound of $\overline{S}(\mathbf{X}, y)$ is no looser than that of $\sum_{i \leq K} \exp(-[\underline{g}_\theta(\mathbf{X}, y)]_i)$, and we can conclude that for the final concrete bounds $\overline{S}(\mathbf{X}, y) \leq \sum_{i \leq K} \exp(-[\underline{g}_\theta(\mathbf{X}, y)]_i)$ holds true, and thereby $\log \overline{S}(\mathbf{X}, y) \leq L(-\underline{g}_\theta(\mathbf{X}, y), y)$.

**Remark 1.** *Despite the assumptions involved above, in the implementation, we generally have different concrete bounds $\underline{g}_\theta(\mathbf{X}, y)$ and $\overline{g}_\theta(\mathbf{X}, y)$ for computing $\overline{S}(\mathbf{X}, y)$ with loss fusion (e.g., our IBP+Backward scheme), compared to the case of computing $L(-\underline{g}_\theta(\mathbf{X}, y))$ without loss fusion (e.g., the scheme used in CROWN-IBP [57]). In the former case, $\underline{g}_\theta(\mathbf{X}, y)$ and $\overline{g}_\theta(\mathbf{X}, y)$ are regarded as*

*intermediate bounds and obtained with IBP, while in the later case, $\underline{g}_\theta(\mathbf{X}, y)$ is obtained with LiRPA and $\overline{g}_\theta(\mathbf{X}, y)$ is unused. Therefore, the relaxation on* exp *when using loss fusion may not be strictly tighter than the IBP bound in computing* $L(-\underline{g}_\theta(\mathbf{X}, y))$.

## C  Additional Details on Experiments

### C.1  Details on Large-Scale Certified Defense

**Training settings**  In order to perform fair comparable experiments, for all experiments on training large-scale vision models (Table 2 and 4), we use a same setting for LiRPA and IBP. Across all datasets, the networks were trained using the Adam [23] optimizer with an initial learning rate of $5 \times 10^{-4}$. Also, gradient clipping with a maximum $\ell_2$ norm of 8 is applied. We gradually increase $\epsilon$ within a fixed epoch length (800 epochs for CIFAR-10, 400 epochs for Tiny-ImageNet and 80 epochs for Downscaled-ImageNet). We uniformly divide the epoch length with a factor $0.4$, and exponentially increase $\epsilon$ during the former interval and linearly increase $\epsilon$ during the latter interval, so that to avoid a sudden growth of $\epsilon$ at the beginning stage. Following [57], for LiRPA training, a hyperparameter $\beta$ to balance LiRPA bounds and IBP bounds for the output layer is set and gradually decreases from 1 to 0 (1 for only using LiRPA bounds and 0 for only using IBP bounds), as per the same schedule of $\epsilon$, and the end $\epsilon$ for training is set to $10\%$ higher than the one in test. All models are trained on 4 Nvidia GTX 1080TI GPUs (44GB GPU memory in total). For different datasets, we further have settings below:

- **CIFAR-10** $\epsilon = \frac{8}{255}$. We train for 2,000 epochs with batch size 256 in total, the first 200 epochs are clean training, then we gradually increase $\epsilon$ per batch with a $\epsilon$ schedule length of 800, finally we conduct 1,100 epochs pure IBP training. We decay the learning rate by $10\times$ at the 1,400-th and 1,700-th epochs respectively. During training, we add random flips and crops for data augmentation, and normalize each image channel, using the channel statistics from the training set.

- **Tiny-ImageNet** $\epsilon = \frac{1}{255}$. We train for 800 epochs with batch size 120 in total (for WideRes-Net, we reduce batch size to 110 due to limited GPU memory), the first 100 epochs are clean training, then we gradually increase $\epsilon$ per batch with a $\epsilon$ schedule length of 400, finally we conduct 500 epochs of pure IBP training. We decay the learning rate by $10\times$ at the 600-th and 700-th epochs respectively. During training, we use random crops of $56 \times 56$ and random flips. During testing, we use a central $56 \times 56$ crop. We also normalize each image channel, using the channel statistics from the training set.

- **Downscaled-ImageNet** $\epsilon = \frac{1}{255}$. We train for 240 epochs with batch size 110 in total, the first 100 epochs are clean training, then we gradually increase $\epsilon$ per batch with a $\epsilon$ schedule length of 80, finally we conduct 60 epochs of pure IBP training. We decay the learning rate by $10\times$ at the 200-th and 220-th epochs respectively. During training, we use random crops of $56 \times 56$ and random flips. During testing, we use a central $56 \times 56$ crop. We also normalize each image channel, using the channel statistics from the training set.

All verified error numbers are evaluated on the test set using IBP with $\epsilon = \frac{8}{255}$ for CIFAR-10 and $\epsilon = \frac{1}{255}$ for Tiny-ImageNet and Downscaled-ImageNet.

**Model Structures**  The details of vision model structures we used are described bellow (note that we omit the final linear layer which has 10 neurons for CIFAR-10 and 200 neurons for Tiny-ImageNet):

- **CNN-7+BN** $5\times$ Conv-BN-ReLU layers with $\{64, 64, 128, 128, 128\}$ filters respectively, and a linear layer with 512 neurons.

- **DenseNet** $\{2, 4, 4\}$ Dense blocks with growth rate 32 and a linear layer with 512 neurons.

- **WideResNet** $3\times$ Wide basic blocks ($6\times$ Conv-ReLU-BN layers) with widen factor = 4 for CIFAR-10, widen factor = 10 for Tiny-ImageNet and Downscaled-ImageNet. An additional linear layer with 512 neurons is added for CIFAR-10.

- **ResNeXt** $\{1, 1, 1\}$ blocks for CIFAR-10 and $\{2, 2, 2\}$ blocks for Tiny-ImageNet and cardinality = 2, bottleneck width = 32 and a linear layer with 512 neurons.

It is worthwhile to mention that both [57] and [58] conducted experiments on expensive 32 TPU cores which has up to 512 GB TPU memory in total. In comparison, our framework with loss fusion can be quite efficient working on 44 GB GPU memory.

Moreover, the running time with maximum batch size on 4 Nvidia GTX 1080TI GPUs of all models on two datasets is reported in Table 8. Note that large-scale models cannot be trained with previous LiRPA methods without loss fusion, even if the mini-batch size on each GPU is only 1 for DenseNet and WideResNet.

Table 8: Per-epoch training time and memory usage of the 4 large models on CIFAR-10 and Tiny-ImageNet with maximum batch size for 4 Nvidia GTX 1080TI GPUs. "LF"=loss fusion. "OOM"= out of memory. Numbers in parentheses are relative to natural training time.

| Data | Training method | Wall clock time (s) | | | | Maximum batch size | | | |
|---|---|---|---|---|---|---|---|---|---|
| | | Natural | IBP | LiRPA w/o LF | LiRPA w/ LF | Natural | IBP | LiRPA w/o LF | LiRPA w/ LF |
| CIFAR-10 | CNN-7+BN | 7.59 | 11.17 (1.54×) | 46.52 (6.13×) | 28.20 (3.71×) | 9500 | 3000 | 600 | 1700 |
| | DenseNet | 9.23 | 37.25 (4.04×) | 187.45 (20.31×) | 74.54 (8.08×) | 2500 | 800 | 150 | 400 |
| | WideResNet | 12.08 | 37.70 (3.12×) | 236.66 (19.59×) | 65.72 (5.44×) | 3000 | 1000 | 160 | 550 |
| | ResNeXt | 6.83 | 19.70 (2.88×) | 130.37 (19.09×) | 43.65 (6.39×) | 4000 | 1200 | 260 | 700 |
| Tiny-ImageNet | CNN-7+BN | 22.17 | 56.54 (2.55×) | 4344.05 (195.94×) | 98.04 (4.42×) | 3600 | 1100 | 12 | 600 |
| | DenseNet | 50.60 | 223.63 (4.42×) | OOM | 474.66 (9.38×) | 800 | 240 | OOM | 120 |
| | WideResNet | 98.01 | 370.68 (3.78×) | OOM | 604.70 (6.17×) | 600 | 200 | OOM | 110 |
| | ResNeXt | 21.52 | 59.42 (2.76×) | 5580.52 (259.32×) | 119.34 (5.55×) | 3200 | 900 | 12 | 500 |

## C.2  Details on Verifying and Training NLP Models

For the perturbation specification defined on synonym-based word substitution, each word $w$ has a substitution set $\mathbb{S}(w)$, such that the actual input word $w' \in \{w\} \cup \mathbb{S}(w)$. We adopt the approach for constructing substitution sets used by Jia et al. [20]. For a word $w$ in a input sentence, they first follow Alzantot et al. [2] to find the nearest 8 neighbors of $w$ in a counter-fitted word embedding space where synonyms are generally close while antonyms are generally far apart. They then apply a language model to only retain substitution words that the log-likelihood of the sentence after word substitution does not decrease by more than 5.0, which is also similar to the approach by Alzantot et al. [2]. We reuse their open-source code[2] to pre-compute the substitution sets of words in all the examples. Note that although we use the same approach for constructing the lists of substitution words as [20], our perturbation space is still different from theirs, because we follow Huang et al. [18] and allow setting a small budget $\delta$ that limits the maximum number of words to be replaced simultaneously [24, 11]. We do not adopt the synonym list from Huang et al. [18] as it appears to be not publicly available when this work is done.

We use two models in the experiments for sentiment classification: Transformer and LSTM. For Transformer, we use a one-layer model, with 4 attention heads, a hidden size of 64, and ReLU activations for feed-forward layers. Following Shi et al. [37], we also remove the variance related terms in layer normalization, which can make Transformer easier to be verified while keeping comparable standard accuracies. For the LSTM, we use a one-layer bidirectional model, with a hidden size of 64. The vocabulary is built from the training data and includes all the words that appear for at least twice. Input tokens to the models are truncated to no longer than 32.

In the certified defense, although we are not using $\ell_p$ norm perturbations, we have an artifial $\epsilon$ that manually shrinks the gap between the clean input and perturbed input during the warmup stage, which makes the objective easier to be optimized [12, 20]. Specifically, for clean input word $w_i$ and actual input word $\hat{w}_i$, we shrink the gap between the embeddings of $w_i$ and $\hat{w}_i$ respectively:

$$e(\hat{w}_i) \leftarrow \epsilon e(\hat{w}_i) + (1 - \epsilon) e(w_i).$$

$\epsilon$ is linearly increased from 0 to 1 during the first 10 warmup epochs. We then train the model for 15 more epochs with $\epsilon = 1$. During the first 20 epochs, all the nodes on the parse trees of training examples are used, and later we only use the root nodes, i.e., the full sentence only. The models are trained using Adam optimizer [23], and the learning rate is set to $10^{-4}$ for Transformer and $10^{-3}$ for LSTM. We also use gradient clipping with a maximum norm of 10.0. When using LiRPA bounds for training, we combine bounds by LiRPA and IBP weighted by a coefficient $\beta (0 \leq \beta \leq 1)$ and $(1 - \beta)$ respectively, and $\beta$ decreases from 1 to 0 during the warmup stage, following CROWN-IBP [57] as also mentioned in Appendix C.1. In this setting, since we use pure IBP for training in the last epochs, we actually end up training the models on $\delta = \infty$ since IBP for LSTM and Transformer does not

consider $\delta$ (see the next paragraph). But we still use LiRPA bounds with the given non-trivial $\delta$ for testing. Alternatively, we also include *IBP+Backward (alt.)* in the experiments, where we always use LiRPA bounds and set $\beta = 1$. And for this setting, the models tend to have a lower verified accuracy when tested on a $\delta$ larger than that in the training, as shown in Sec. 4.

Huang et al. [18] has a convex hull method to handle word replacement with a budget limit $\delta$ in IBP. For a word sequence $w_1, w_2, \cdots, w_l$, they construct a convex hull for the input node 1. They consider the perturbation of each word $w_i$, and for each possible $\hat{w}_i \in \{w_i\} \cup \mathbb{S}(w_i)$, they add vector $[e(w_{1...i-1}); e(w_i) + \delta(e(\hat{w}_i) - e(w_i)); e(w_{i+1...l})]$ to the convex hull. The convex hull is an over-estimation of $h_1(\mathbf{X})$. They require the first layer of the network to be an affine layer and concretize the convex hull to interval bounds after passing the first layer, where each vertex in the convex hull is passed through the first layer respectively and they then take the interval lower and upper bound of all the vertexes in the convex hull. They worked on CNN, but on Transformer where the first layer is a linear layer independently applied to each position in the sequence, their method is a $(\delta - 1)$-time more over-estimation than simply assuming all the words can be replaced at the same time, and this method cannot work either when the first layer is not an affine layer. Therefore, for verifying and training LSTM and Transformer with IBP, we adopt the baseline in Jia et al. [20] without considering $\delta$. In contrast, our dynamic programming method for concretizing linear bounds under the synonym-based word substitution scenario in Sec. 3.2 is able to consider the budget $\delta$ regardless of the network structure.

## C.3 Details on Training for a Flat Objective

**Hyperparameter Setting**  For training the three-layer MLP model we used in weight perturbation experiments, we follow similar training strategy in vision models. The differences are summarized here: We use the SGD optimizer with an initial learning rate of $0.1$ and decay the learning rate with a factor of $0.5$ after $\epsilon$ increases. We use $\ell_2$ norm with $\epsilon = 0.1$ to bound the weights of all three layers and linearly increase $\epsilon$ per batch.

**Certified Flatness**  Using bounds obtained from LiRPA, we can obtain a certified upper bound on training loss. We define the flatness based on certified training cross entropy loss at a point $\theta^* = [\mathbf{w}_1^*, \mathbf{w}_2^*, \cdots, \mathbf{w}_K^*]$ as:

$$\mathcal{F} = \mathcal{L}(-\underline{\mathbf{h}}(\mathbf{x}, \theta^*, \epsilon); y) - \mathcal{L}(\mathbf{h}(\mathbf{x}, \theta^*); y) \geq \max_{\mathbf{w} \in \mathbb{S}} \mathcal{L}(\theta) - \mathcal{L}(\theta^*). \tag{17}$$

A small $\mathcal{F}$ guarantees that $\mathcal{L}$ does not change wildly around $\theta^*$. Note that since the weight of each layer can be in quite different scales, we use a normalized $\bar{\epsilon} = 0.01$ and set $\epsilon_i = \|\mathbf{w}_i\|_2 \bar{\epsilon}$. This also allows us to make fair comparisons between models with weights in different scales. The flatness $\mathcal{F}$ of the models we obtained are shown in Table 9. As we can see, the models trained by "flat" objective show extraordinarily smaller flatness $\mathcal{F}$ compare with the naturally trained models on both MNIST and FashionMNIST with all combination of dataset sizes and batch sizes. The results also fit the observation of training loss landscape in Figure 3b.

Table 9: The flatness $\mathcal{F}$ of naturally trained models and models trained using the "flat" objective (17) with different dataset sizes (10%, 1%) and batch sizes ($0.01N$, $0.1N$, $N$). A small $\mathcal{F}$ guarantees that $\mathcal{L}$ does not change wildly around $\theta^*$ (model parameters found by SGD). The flat objective provably reduces the range of objective around $\theta^*$.

| | MNIST | | | | | |
|---|---|---|---|---|---|---|
| | natural training | | | "flat" objective | | |
| | $0.01N$ | $0.1N$ | $N$ | $0.01N$ | $0.1N$ | $N$ |
| 10% | 2.79 | 3.45 | 4.55 | 0.97 | 1.12 | 1.83 |
| 1% | 2.96 | 3.85 | 4.77 | 1.10 | 0.95 | 1.44 |
| | FashionMNIST | | | | | |
| 10% | 7.89 | 7.95 | 9.60 | 2.49 | 1.81 | 1.94 |
| 1% | 7.86 | 6.43 | 9.55 | 2.52 | 1.79 | 1.98 |

## C.4 Details on Verification and Training under $\ell_0$-norm Perturbation

**Concretization**  We can handle any input constraint $\mathbf{X} \in \mathbb{S}$ as long as the linear "concretization" problem (6) can be efficiently solved. When $\mathbb{S}$ is an $\ell_\infty$ ball, the problem is linear, and when $\mathbb{S}$ is an

Table 10: Results of $\ell_0$ norm certified defense on MNIST.

| Method | Metric | k = 1 | k = 4 | k = 10 |
|--------|--------|-------|-------|--------|
| IBP | Clean err. | 1.57% | 2.24% | 4.84% |
| | Verified err. | 5.79% | 10.06% | 25.15% |
| Ours | Clean err. | 1.62% | 2.21% | 4.95% |
| | Verified err. | **5.71%** | **9.59%** | **24.67%** |

$\ell_2$ ball, this is non-linear but is still convex and therefore relatively easy to solve. Now we will show that we can even handle non-linear, non-convex cases. For example, when $\mathbb{S}$ is a sparse perturbation, e.g, an $\ell_0$ ball: $\mathbb{S} = \{\|\mathbf{X} - \mathbf{X}_0\|_0 \leq k, 0 \leq \mathbf{X} \leq 1\}$, the optimization problem (6) is to find the $k$ pixels that can change the output most when perturbed to the boundary 0 or 1. Since the optimization problem is linear and does not have correlations between input pixels, we can simply get the solution by ranking the changes caused by perturbing each individual pixel and choose the top-k of them:

$$
\begin{aligned}
\underline{\mathbf{h}}_{o,j} &= \mathbf{A}_{j,:}^{\mathrm{T}}\mathbf{X} - \sum_{i \in \underline{K}_j} \left(\mathbf{A}_{j,i}^{+} \cdot \mathbf{X}_i - \mathbf{A}_{j,i}^{-} \cdot (1 - \mathbf{X}_i)\right) \\
\text{where } \underline{K}_j &:= \arg\underset{i}{\text{top-k}} \left(\mathbf{A}_{j,i}^{+} \cdot \mathbf{X}_i - \mathbf{A}_{j,i}^{-} \cdot (1 - \mathbf{X}_i)\right) \\
\overline{\mathbf{h}}_{o,j} &= \mathbf{A}_{j,:}^{\mathrm{T}}\mathbf{X} + \sum_{i \in \overline{K}_j} \left(-\mathbf{A}_{j,i}^{-} \cdot \mathbf{X}_i + \mathbf{A}_{j,i}^{+} \cdot (1 - \mathbf{X}_i)\right) \\
\text{where } \overline{K}_j &:= \arg\underset{i}{\text{top-k}} \left(-\mathbf{A}_{j,i}^{-} \cdot \mathbf{X}_i + \mathbf{A}_{j,i}^{+} \cdot (1 - \mathbf{X}_i)\right),
\end{aligned}
\tag{18}
$$

where $\arg\text{top-k}_i$ denotes the set of indices of the largest $k$ elements among all indices of $i$, $\mathbf{A}^{+} = \max(0, \mathbf{A})$, $\mathbf{A}^{-} = \min(0, \mathbf{A})$ are positive and negative elements of matrix $\mathbf{A}$.

After the concretization step and obtaining the bounds, we can apply verification and certified training similar to the $\ell_\infty$ and $\ell_2$ cases.

**Training** We show preliminary results on LiRPA based $\ell_0$ norm certified defense in Table 10. In experiments we follow the settings in Chiang et al. [4] and trained a simple MLP model with three hidden layers with [256, 256, 128] neurons on MNIST dataset. Our reproduced IBP results match those reported in Chiang et al. [4], and our LiRPA based certified defense achieves consistent improvements compared to IBP.

## Footnotes

[2]`https://bit.ly/2KVxIFN`