[Reviews · NeurIPS 2020]

Review 1

Summary and Contributions: This paper proposes perturbation analysis on general computational graph structures. The analysis is based on LiPRA algorithms, which analyze the tight lower bound and upper bound. The evaluation on both image feed-forward NN case, and NLP LSTM and Transformer models demonstrate the potentiality of proposed methods.

Strengths: + This paper works on an important problem. + The proposed technique should be a feasible solution. + Important contribution for general graph structure analysis with loss fusion for scalability + The evaluation covers diverse model architecture and tasks.

Weaknesses: - For the image and feed-forward NN case, existing methods such as abstract interpretation and other methods are also applicable, which should be compared for bound tightness analysis and efficiency. - The evaluations are performed on small datasets. But I do not see this to be a severe issue for formal bound analysis.

Correctness: The proposed techniques are sound and properly evaluated, although more comparison with existing methods are highly desiable.

Clarity: The paper is well written, dense, but easy to follow

Relation to Prior Work: Mostly

Reproducibility: Yes

Additional Feedback: Please refer to the comments. Post-rebutal: The authors properly address my concerns in the rebutal phase. This is a solid work and important contribution, which I would raise my score.


Review 2

Summary and Contributions: In this work, the authors create a method for doing linear bounding of general computational graphs. Their primary contribution is the algorithm for establishing and propagating linear bounds through computation graphs that may have irregular structure. This has the advantage of this framework is that it allows for perturbation analysis without hand-coding linear bounds. The authors then use their tool on a variety of tasks and evaluate its performance both with the verifications the tool is able to provide as well as the memory and computational requirements of their methods.

Strengths: I think this work definitely makes a solid contribution to the NeurIPS community as perturbation analysis is an important tool for furthering our theoretical and empirical understanding of the working of deep learning systems. The paper is well presented and has minimal type-os The concept is pretty easy to follow and the algorithm is clean enough to be read and understood. They provide a good covering in terms of tool evaluation. The authors evaluate their tool on "benchmarks" by which I mean present their tools results on datasets and models which have been done before to show that their method works. Further, they add to discourse, albeit in a minor way, with their analysis of loss landscapes to show how the tool can be used to test current hypotheses in the deep learning literature.

Weaknesses: There are two handicaps that I can see to this work: firstly, the methodology while impressive in its ability to generalize to LSTMs, CNNs, and Transformers actually seems like it may be limited in terms of its scope of verification procedures it admits. In the abstract the authors mention that they can implement things such as CROWN into their LiRPA algorithm, however CROWN also explicitly derives quadratic over and under approximations of pre-activation bounds and I am curious as to if the authors have a scheme for directly accommodating things of this nature along with higher-order convex relaxations (e.g. even degree 3 polynomials for small networks) which are harder to propagate in general as there are more complex dependencies. [I recognize that this may be difficult to do.] Further on this point, the method can really only handle linear constraints in the input. Indeed, the authors show how they can bound discrete perturbations in the NLP scenario, but at the end, it looks to me like the authors recursively build a linear region in the input space by taking maximally perturbed embeddings from the synonym set. Is there a plan for how to incorporate non-linear specifications? I suppose if the specification is non-linear but monotonic all of the same propositions hold, but I think it would be nice if the authors clearly laid out a plan for future work and extending their tool. Even sketches of how these computations would be compatible with the current framework would contribute, in my mind, to the potential significance of the method. The second potential weakness I see exist in the experimental validation of the framework. Unfortunately, it doesn't tell us anything particularly new. The authors compare their tool with IBP, but it isn't a really fair comparison as IBP is inherently more approximate than any linear bound propagation. Finally, I am hoping the authors may be able to find some space to address the kind of odd performance on the NLP benchmark. I have read other papers which do LBP for NLP models ([15] in the paper), and while I recognize the difference in setting, in that work there was a clear degradation of model performance (in terms of verifiability) as the perturbation magnitude increased. Which is what I would expect to be true, largely, of all neural network classifiers. Yet, this method reports that verified error remains constant across all delta's? At least in the rebuttal can the authors give me some intuition for why this is the case? I just hope it is not indicative of a bug in the code is all. **************************** After author response: I would like to thank the authors for their clear and concise answers to my questions and confusions. I think this is a solid and mature work that makes an interesting and important contribution to those interested in checking the provable robustness of their models.

Correctness: To the best of my knowledge all of the proofs seem sound and I have read all of the formulations and they seem correct to me. There is a type-o (I think) just before equation (3) where the authors define two lower bounds on A_i when I think they meant to define a lower and upper bound.

Clarity: Yes the paper was very clearly written.

Relation to Prior Work: I think it is clear that the authors rely greatly on the previously proposed methods of linear relaxation to get bounds but make their contribution in the framework which can be applied to general computational graphs.

Reproducibility: Yes

Additional Feedback: One thing I would suggest to the authors as a clear area that they could contribute and gain novel insights is in Bayesian Neural Networks. The formulation of probabilistic safety for BNNs was only recently published in UAI (https://arxiv.org/abs/2004.10281) and it seems the authors do not do LBP for their BNNs on MNIST and do little analysis of BNNs at the scale that this paper considers. I think this is a great place for an application of this method where the authors can show clear improvement over state of the art, especially given that their tool (in the flatness section) shows that they can handle constraints in the parameter space. Perhaps even including this in future aims would be nice.


Review 3

Summary and Contributions: This paper proposes a framework to enable LiRPA for a wide range of neural networks, including DenseNet, ResNeXt and Transformers. Existing work either do not scale to these complex model architectures, or derives architecture-specific bounds for verification. To my best knowledge, this is the first work to unify and scale LiRPA-based neural network verification to these more advanced model architectures. Specifically, their framework computes the lower and upper bounds with both the forward and backward modes, and they develop a library to automatically propagate the bounds, similar in spirit to backpropagation for neural network training. Meanwhile, they propose the loss fusion technique, which reduces the time complexity for bound computation by a factor of the label set size. Therefore, they can scale the robust training to datasets with much larger label sets (i.e., Tiny ImageNet with 200 labels) than studied in prior work on certified defense. Lastly, their approach can be extended for verification beyond L_p perturbations. For example, they also consider synonym-based word substitution for NLP models. They evaluate several model architectures on CIFAR-10 and Tiny ImageNet for image classification, and SST-2 for sentiment classification. They show that their framework achieves better nature error rate and verified error rate than existing verification approaches. Meanwhile, the loss fusion technique significantly reduces the training time and memory usage.

Strengths: 1. To my best knowledge, this is the first work to unify and scale LiRPA-based neural network verification to multiple complex model architectures, ranging from vision to NLP models. 2. The proposed loss fusion technique significantly reduces the training time and memory usage, and scales training to support dataset with a larger label set than those considered in previous certifiable defense work.

Weaknesses: 1. From the Appendix, both Transformer and LSTM in this paper only include one layer. Does the approach still work for multi-layer NLP models? 2. Does "nature error rate" mean the error rate on the clean test set? If so, I feel that the error rates are too low compared to models trained in a standard way. Some explanation could be helpful. UPDATE: I thank the authors for addressing my questions, and I keep my original review score.

Correctness: The approach is technically sound.

Clarity: The paper is generally clear. However, the authors should do a careful proofreading to fix some typos. For example: Line 27 in the Introduction, the terms should be f(x0+δ) instead of f(δ). Line 54~56: "LiRPA" is misspelled as "LiPRA". Line 255: "Transfomrer" -> "Transformer".

Relation to Prior Work: The paper provides a good comparison with existing work.

Reproducibility: Yes

Additional Feedback:

[Author Response · NeurIPS 2020]

Table A: Verified error and running time of different frameworks on CIFAR. The model structure is ConvSmall from [28].

| | Regularly trained ($\epsilon$=2/255) | | | LiRPA trained ($\epsilon$=8/255) | | | |
| --- | --- | --- | --- | --- | --- | --- | --- |
| | [35] | [37] | Ours | [45] | [28] | [50] | Ours |
| Verified error | 63.85% | 63.85% | 63.85% | 74.50% | 75.30% | 71.59% | **71.57%** |
| Time (min) | 223.37 | 89.45 | **14.37** | 248.74 | 105.31 | 43.45 | **28.11** |

Table B: Certified training on Downscaled ImageNet. We use WideResNet with $\epsilon = \frac{1}{255}$.

| Dataset | Method | Clean | PGD | Verified |
| --- | --- | --- | --- | --- |
| ImageNet ($64 \times 64$) | IBP [9] | 84.04% | 90.88% | 93.87% |
| | Ours | **83.77%** | **89.74%** | **91.27%** |

We thank all reviewers for their encouraging and helpful comments. We will fix all typos. We answer questions below:
**R1.** Comparison to other LiRPA implementations on feed-forward NNs. We categorize existing implementations
into 2 kinds: (1) for verification only (typically implemented on CPUs, including DeepZ[35], and DeepPoly[37])
(2) for training certified defense (typically using more efficient, yet weaker or approximated bounds: convex outer
adversarial polytope[45], DiffAI[28], IBP[9] and CROWN-IBP[50]). For category (1) we compare bound tightness
(verified error given a $\ell_\infty$ norm $\epsilon$) and time to verify the test set; for category (2) we compare verified accuracy *after*
*training* and training time. Results are presented in Table A. Following convex relaxation theory[32], our bound has
the same strength as CROWN[49]/DeepPoly[37], but we use GPU acceleration from PyTorch. Our contribution is not
to improve tightness of LiRPA bounds, but the first framework that generalizes to general computational graphs in
an automatic manner. Results on a large dataset. We conduct additional experiments on downscaled ($64 \times 64$, 1,000
classes) ImageNet in Table B. With the help of loss fusion, for the first time, we demonstrate LiRPA based certified
defense on Downscaled ImageNet and outperform IBP[9], the only method that can scale to this setting previously.

**R2.** High-order bounds. Admittedly, we only implement the linear relaxations of CROWN and currently do not handle
CROWN-quad. In CROWN[50], the quadratic bound is only applied to 2-layer networks and is hard to extend to
multiple layers, as when propagating a quadratic bound to the 3rd layer it becomes quadratic ($x^4$) due to correlations
between two quadratic terms ("order explosion"). This makes the concretization problem (in Sec. 3.2) hard to solve.
We plan to study high-order bounds on general graphs as our future work. Limitations on linear input constraints We
can handle any input constraint $\mathbf{X} \in \mathbb{S}$ as long as the linear "concretization" problem can be efficiently solved (Sec 3.2).
When $\mathbb{S}$ is an $\ell_\infty$ ball, it is linear; but it is non-linear when $\mathbb{S}$ is an $\ell_2$ ball (but $\mathbb{S}$ is still convex so easy to solve). We can
even handle non-linear, non-convex case. For example, when $\mathbb{S}$ is a sparse perturbation (non-linear and non-convex),
e.g, $\mathbb{S} = \{\|\mathbf{X} - \mathbf{X}_0\|_0 \leq k, 0 \leq \mathbf{X} \leq 1\}$, the solution is: $\underline{\mathbf{h}}_{o,j} = \mathbf{A}_{j,:}\mathbf{X} - \sum_{\text{topk}}(\mathbf{A}_{j,:}^+ * \mathbf{X}) + \sum_{\text{topk}}(\mathbf{A}_{j,:}^- * (1 - \mathbf{X}))$,
$\overline{\mathbf{h}}_{o,j} = \mathbf{A}_{j,:}\mathbf{X} - \sum_{\text{topk}}(\mathbf{A}_{j,:}^- * \mathbf{X}) + \sum_{\text{topk}}(\mathbf{A}_{j,:}^+ * (1 - \mathbf{X}))$. where $*$ denotes element-wise multiplication, $_\text{topk}$ denotes the
indices of largest $k$ elements. We show preliminary results on LiRPA based $\ell_0$ norm certified defense in Table D. The
input constraints can be even more generalized when it is produced by some parameterized neural network, where we
can combine this network with the classifier to verify the whole computation. We will also discuss these extensions.
Fairness of comparison to IBP. We compare to IBP in training experiments because (1) IBP is currently the only feasible
method for training large-scale certified defense on irregular networks; (2) Even on smaller networks, IBP outperforms
many tighter bounds *after training* (see Table 4 in [9]) and IBP based method[50] is currently the state-of-the-art.
Performance on NLP benchmarks We discussed this issue in Appendix C.2 (L.545-559). Huang et al. build a convex
hull on the input layer, where each instance in the convex hull has only one position perturbed but the perturbation
is enlarged to $\delta$ times. They use CNN and after the first layer, the convex hull is converted into interval bounds. But
this requires the first layer to be an affine layer and different positions have interactions, which is not the case in
Transformer/LSTM. E.g., the linear layer before the self-attention in Transformer (to obtain query/key/value) is applied
to different positions independently. In this case, Huang et al.'s method gives a $(\delta - 1)$-time over-estimation, compared
to assuming all the positions are independently replaced as in Jia et al. [17]. Therefore, we adopt the IBP based method
in [17] whose results are not affected by $\delta$. To avoid bugs, we test our code base carefully with continuous integration
(Travis CI) and we compare our bounds with references from other libraries when possible (e.g., on feed-forward NNs).
Bayesian Neural Networks We greatly appreciate the reviewer on pointing out this potential application and we will
discuss it in related works and further study it as our future work.

Table C: Multi-layer NLP models with $\delta_{\text{train}} = 6$.

| Model | Method | Verified Test Accuracy (%) | | | |
| --- | --- | --- | --- | --- | --- |
| | | $\delta = 0$ | $\delta = 1$ | $\delta = 3$ | $\delta = 6$ |
| 2-Layer Transformer | IBP | 77.5 | 75.4 | 75.4 | 75.4 |
| | Ours | 78.1 | 77.2 | 77.2 | 77.2 |
| 4-Layer Transformer | IBP | 78.4 | 76.0 | 76.0 | 76.0 |
| | Ours | 78.6 | 77.4 | 77.4 | 77.3 |
| 2-Layer LSTM | IBP | 81.4 | 78.2 | 78.2 | 78.2 |
| | Ours | 81.4 | 78.4 | 78.4 | 78.4 |

Table D: Results of $\ell_0$ norm certified defense on a simple MLP model.

| Method | Metric | k = 1 | k = 4 | k = 10 |
| --- | --- | --- | --- | --- |
| IBP | Verfied err. | 5.79% | 10.06% | 25.15% |
| | Clean err. | 1.57% | 2.24% | 4.84% |
| Ours | Verfied err. | 5.71% | 9.59% | 24.67% |
| | Clean err. | 1.62% | 2.21% | 4.95% |

**R3.** Multi-layer NLP models. Our method natively support multi-layer LSTMs and Transformers. We include additional
experiments in Table C. Our framework provides competitive results on these networks. Nature error rates Yes, it is the
error rate on clean test set (we will fix the terms used). The accuracy of models is relatively low compared to normally
trained models, but this is common in certified defense - our reported error rates are similar to or better than those in
state-of-the-art (e.g., [45,9,50] reported *clean test errors* of 71.33%, 50.51% and 54.02% on CIFAR-10 with $\epsilon = \frac{8}{255}$,
respectively; ours are around 53%). Currently all LiRPA based certified defenses have this trade-off between robustness
and accuracy. We leave further development on improving the clean accuracy of certified training as a future work.

[Meta-Review · NeurIPS 2020]

The authors agree that this paper presents a nice tool that generalizes the IBP method for certifying simple neural nets.